# EVENT-DRIVEN ONLINE VERTICAL FEDERATED LEARNING

**Ganyu Wang**[1]    **Boyu Wang**[1,2]    **Bin Gu**[3*]    **Charles Ling**[1,2*]
[1]Western University    [2]Vector Institute    [3]Jilin University
gwang382@uwo.ca    bwang@csd.uwo.ca    jsgubin@gmail.com
charles.ling@uwo.ca

## ABSTRACT

Online learning is more adaptable to real-world scenarios in Vertical Federated Learning (VFL) compared to offline learning. However, integrating online learning into VFL presents challenges due to the unique nature of VFL, where clients possess non-intersecting feature sets for the same sample. In real-world scenarios, the clients may not receive data streaming for the disjoint features for the same entity synchronously. Instead, the data are typically generated by an *event* relevant to only a subset of clients. We are the first to identify these challenges in online VFL, which have been overlooked by previous research. To address these challenges, we proposed an event-driven online VFL framework. In this framework, only a subset of clients were activated during each event, while the remaining clients passively collaborated in the learning process. Furthermore, we incorporated *dynamic local regret (DLR)* into VFL to address the challenges posed by online learning problems with non-convex models within a non-stationary environment. We conducted a comprehensive regret analysis of our proposed framework, specifically examining the DLR under non-convex conditions with event-driven online VFL. Extensive experiments demonstrated that our proposed framework was more stable than the existing online VFL framework under non-stationary data conditions while also significantly reducing communication and computation costs.

## 1 INTRODUCTION

Vertical Federated Learning (VFL) (Vepakomma et al., 2018; Yang et al., 2019; Liu et al., 2019; Chen et al., 2020; Gu et al., 2020; Zhang et al., 2021b;a; Wang et al., 2023; Qi et al., 2022; Wang et al., 2024) is a privacy-preserving machine learning paradigm wherein multiple entities collaborate to construct a model without sharing their raw data. In VFL, each participant possesses non-intersecting features for the same set of samples, which is significantly different from the Horizontal Federated Learning (HFL) (McMahan et al., 2017; Karimireddy et al., 2020; Li et al., 2020; 2021; Marfoq et al., 2022; Mishchenko et al., 2019) where each client possesses the non-overlap samples of the same features.[1]

Current research on VFL primarily focuses on the offline scenario, characterized by a pre-established dataset. However, the limitations of offline learning become obvious when building real-world applications of VFL. First, offline learning is unsuitable for scenarios where the dataset undergoes continual updates, which is typical in real-world applications. For example, in the application scenario of VFL scenarios involving companies as clients (Wei et al., 2022; Vepakomma et al., 2018), new data is constantly generated as new customers engage with the companies, or as existing customers update their records through ongoing activities. Similarly, in VFL scenarios involving edge devices (Wang & Xu, 2023; Liu et al., 2022), the sensors, acting as the clients, continuously receive data streams from the environment rather than maintaining a static dataset. Second, the dynamic nature of real-world environments leads to data distribution drift, which is particularly evident in edge devices. In response, the offline learning paradigm requires retraining the model from scratch to accommodate shifts in data distribution. While this retraining process may not present significant

---

*Corresponding authors.
[1]The term Federated Learning commonly refers to HFL. However, this is not the setting of this study.

challenges in centralized learning, it becomes prohibitively expensive in distributed learning scenarios, as the training process imposes substantial communication and computation costs in distributed learning. Consequently, online learning may provide greater adaptability in VFL by allowing models to update continuously as new data arrives and handle dynamic environments.

However, applying online learning to VFL is not straightforward due to its inherent nature. First, in online VFL, clients receive non-intersecting features of the data from the environment. In real-world scenarios, it is rare for all clients to receive all these features of a sample simultaneously. Instead, it is more common for only a subset of clients to obtain the relevant features in response to a specific *event*. For example, in VFL implementations within large companies (Wei et al., 2022; Hu et al., 2019; Vepakomma et al., 2018), when a customer takes an action such as making a payment or a purchase, this action typically involves only one company of the VFL, while the data from the other companies remain unchanged. Similarly, in VFL involving sensor networks (Wang & Xu, 2023; Liu et al., 2022), only the sensor triggered by an event will be activated, while others remain inactive (Suh, 2007; Heemels et al., 2012; Trimpe & D'Andrea, 2014; Beuchert et al., 2020). The above scenario has brought about the demand for an event-driven online VFL framework, wherein certain participants are activated by events during each round, thereby dominating the learning process, while the rest of the participants remain inactive or passively cooperate with the learning.

The second challenge in online VFL lies in addressing non-convex models and dynamic environments, which are prevalent in practical applications. Current research in online VFL still primarily focuses on online convex optimization, assuming both convex models and stationary data streams (Wang & Xu, 2023). While convex models are easier to optimize, they fail to capture the complex patterns necessary for tackling more challenging tasks. Additionally, the assumption of stationary data streams is unrealistic in dynamic real-world environments, where data distributions can shift over time. For example, the environmental sensor that monitors the air quality may experience dynamic change due to natural phenomena. These are challenges that current online VFL frameworks have yet to resolve.

To address the aforementioned challenges, we propose a novel event-driven online VFL framework, which is well-suited for the online learning scenario in non-convex cases and non-stationary environments. Figure 1 depicts a schematic graph of our framework. In our framework, a subset of the clients are activated by the event at each round, while others passively contribute to the training. This approach substantially reduces communication-computation costs and facilitates the client model in learning relevant content. Moreover, we adapt the dynamic local regret approach proposed by Aydore et al. (2019) to our event-driven online VFL framework to effectively handle online learning in non-convex cases and non-stationary environments.

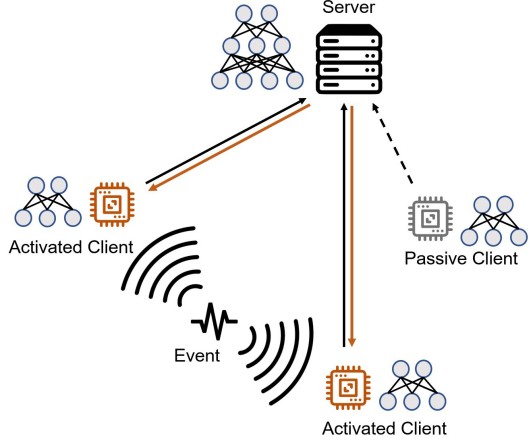

Figure 1: Event-driven online VFL

In summary, the contributions of our paper are:

- We identify the unrealistic assumption of synchronous data reception in online VFL research and propose a novel event-driven online VFL paradigm that is better suited to real-world scenarios.

- We adapt the dynamic local regret approach to our event-driven online VFL to effectively handle non-convex models in non-stationary data streaming scenarios, and we theoretically prove the dynamic local regret bound for this framework, which incorporates partial activation of the client.

- Our experiments demonstrate that our event-driven online VFL exhibits greater stability compared to existing methods when confronted with non-stationary data conditions. Additionally, it significantly reduces communication and computation costs.

**Notation** We use a square bracket with multiple items to denote concatenation for convenience. For instance, given $w_A \in \mathbb{R}^{d_A}$, $w_B \in \mathbb{R}^{d_B}$, we define $[w_A, w_B] \triangleq [w_A^\top, w_B^\top]^\top$. The superscript $t$

attached to the parameter $w$ indicates the number of rounds (time step). A square bracket enclosing a single integer represents the set of natural numbers from 1 to that particular number. For instance, $[M] = \{1, 2, \ldots, M\}$.

## 2   RELATED WORK: ONLINE HFL AND ONLINE VFL[2]

**Online HFL**   Most existing online federated learning research focuses on HFL because extending the HFL framework to online learning is relatively straightforward. In HFL, each participant possesses the complete feature set of the local sample. Therefore, online HFL can be easily achieved by assigning each client a unique data stream containing non-overlapping samples. The current research on online HFL is focused on speeding up optimization (Mitra et al., 2021; Eshraghi & Liang, 2020), reducing communication cost (Hong & Chae, 2021) and dealing with concept drift (Ganguly & Aggarwal, 2023). Mitra et al. (2021) applied online mirror descent within the Federated Learning framework, demonstrating sub-linear regret in convex scenarios. Hong & Chae (2021) introduced a randomized multi-kernel algorithm for online federated learning, which maintains the performance of the multi-kernel algorithm while mitigating the linearly increasing communication cost. Kwon et al. (2023) incorporated client sampling and quantization into online federated learning. Ganguly & Aggarwal (2023) proposed a non-stationary detection and restart algorithm for online federated learning, addressing the concept of drift during online learning.

**Online VFL**   In VFL, each client possesses a non-overlapping feature set, which presents challenges when integrating online learning into this framework. The existing approach to Online VFL is proposed by Wang & Xu (2023), which applies online convex optimization to synchronous VFL. However, they naïvely assume that all clients receive a synchronous data stream, which does not align with real-world applications. Apart from this study, no other research has explored online VFL and the characteristics of dataset streaming in this context. Through the exploration of event-driven mechanisms, we open up new possibilities for real-time data streaming processing across distributed nodes within the VFL framework.

## 3   METHOD

### 3.1   PROBLEM DEFINITION

In the VFL framework, there is a single server and $M$ clients. The server produces the label $y^t$ at round $t$, while each client may receive non-intersecting features $x_m^t$ from the environment during the same round. The model for client $m$, denoted as $h_m(w_m; x_m^t)$, is parameterized by $w_m \in \mathbb{R}^{d_m}$ and takes the local feature $x_m^t$ as input to produce an embedding. We denote $\mathbf{w}$ as the concatenation of the parameter from all clients, i.e. $\mathbf{w} = [w_1, \cdots, w_M]$. The server, parameterized by $w_0$, recieves embeddings $h_m(\cdot)$ from all clients and then calculates the losses with the label $y^t$. We define the VFL framework in composite form.

$$f^t(w_0, \mathbf{w}, x^t; y^t) = f\left(w_0, h_1(w_1; x_1^t), \cdots, h_M(w_M; x_M^t); y^t\right) \tag{1}$$

where $f(\cdot)$ denotes the model on the server. For brevity, we denote $f^t(w_0, \mathbf{w}, x^t; y^t)$ by $f^t(w_0, \mathbf{w})$ throughout all following sections.

Following the work from Aydore et al. (2019), we employ dynamic local regret analysis in online non-convex optimization. To reformulate the dynamic local regret under the context of the VFL framework, we begin by introducing the concept of exponentially weighted sliding-window average as the basis for its computation.

**Definition 1.** *Exponential weighted sliding-window average: Let $w_0^t \in \mathbb{R}^{d_0}$ denote the server's parameter at time $t$, $w_m^t \in \mathbb{R}^{d_m}$ be the client $m$'s parameter at time $t$. $\mathbf{w} = [w_1, w_2 \cdots w_M]$. Let $l$ denote the length of the sliding window. Then the exponential weighted sliding-window average can be defined as follows:*

$$S_{t,l,\alpha}(w_0^t, \mathbf{w}^t) \triangleq \frac{1}{W} \sum_{i=0}^{l-1} \alpha^i f^{t-i}(w_0^{t-i}, \mathbf{w}^{t-i}) \tag{2}$$

---

[2]Extra discussion on the related work of VFL and online learning is in Appendix E.

*where $0 < \alpha < 1$ and the superscript $i$ of the $\alpha^i$ indicates the exponent. $W = \sum_{i=0}^{l-1} \alpha^i$ serves as the normalization parameter for the exponential average, ensuring that $\frac{1}{W} \sum_{i=0}^{l-1} \alpha^i = 1$. It is worth noting that this window gives more weight to recent values, with the weight decaying exponentially, and the loss for $f^{t-i}(w_0^{t-i}, \mathbf{w}^{t-i})$ is computed on the past parameter at round $t-i$.*

Then, the DLR can be formally defined based on the accumulated square norm of the gradient of the exponentially weighted sliding-window average.

**Definition 2.** *Dynamic $l$-local regret: Let $S_{t,l,\alpha}(w_0^t, \mathbf{w}^t)$ be the sliding-window defined above, $w_0$ be the server's parameter and $\mathbf{w}$ be the aggregated clients' parameter. The Dynamic $l$-Local Regret can be defined as:*

$$DLR_l(T) \triangleq \sum_{t=1}^{T} \left\| \nabla S_{t,l,\alpha}(w_0^t, \mathbf{w}^t) \right\|^2 \tag{3}$$

## 3.2 ADAPT DLR TO ONLINE VFL

We integrate the dynamic exponentially time-smoothed online gradient descent method introduced by Aydore et al. (2019) into the VFL framework by incorporating a buffer to store the past gradients.

**Server update** Based on the special characteristic of the dynamic local regret, the server is required to maintain a buffer of the past intermediate derivative values of length $l$. At each time step, the server computes the gradient and updates the buffer by enqueuing the latest gradient and dequeuing the oldest. Subsequently, the server utilizes the buffer to compute the dynamic exponentially time-smoothed gradient. Specifically, the partial derivative w.r.t. the server is shown in Eq. 4, and the buffer stores $l$ past gradients.

$$\nabla_{w_0} S_{t,l,\alpha}(w_0^t, \mathbf{w}^t) = \frac{1}{W} \sum_{i=0}^{l-1} \underbrace{\alpha^i \nabla_{w_0} f^{t-i}(w_0^{t-i}, \mathbf{w}^{t-i})}_{\text{Server Buffer, } i = 0, 1, \cdots l-1} \tag{4}$$

Finally, the server updated its model with stochastic gradient descent, i.e. $w_0^{t+1} \leftarrow w_0^t - \eta_0 \cdot \nabla_{w_0} S_{t,l,\alpha}(w_0^t, \mathbf{w}^t)$, where $\eta_0$ is the learning rate for the server.

**Client update** The client cannot calculate the partial derivative w.r.t. its model by themselves because they do not hold the label. Consequently, they depend on the server to transmit the partial derivative $v_m^t = \frac{\partial f^t(w_0^t, \mathbf{w}^t)}{\partial h_m(w_m^t; x_m^t)}$ w.r.t. the client's model output $h_m$ to facilitate model updates. The partial derivative w.r.t. the client $m$'s model is computed through chain rules:

$$\nabla_{w_m} \tilde{S}_{t,l,\alpha}(w_0^t, \mathbf{w}^t) = \frac{1}{W} \sum_{i=0}^{l-1} \alpha^i \nabla_{w_m} f^{t-i}(w_0^{t-i}, \mathbf{w}^{t-i})$$

$$= \frac{1}{W} \sum_{i=0}^{l-1} \alpha^i \underbrace{\frac{\partial f^{t-i}(w_0^{t-i}, \mathbf{w}^{t-i})}{\partial h_m(w_m^{t-i}; x_m^{t-i})} \cdot \frac{\partial h_m(w_m^{t-i}; x_m^{t-i})}{\partial w_m^{t-i}}}_{\text{Client Buffer, } i = 0, 1 \cdots l-1} \tag{5}$$

After receiving $v_m^t$ from the server, the clients update their buffer by enqueuing the $v_m^t \cdot \frac{\partial h_m(w_m^t; x_m^t)}{\partial w_m^t}$. Finally, the client is updated with $w_m^{t+1} \leftarrow w_m^t - \eta_m \nabla_{w_m} S_{t,l,\alpha}(w_0^t, \mathbf{w}^t)$.

## 3.3 EVENT-DRIVEN ONLINE VFL FRAMEWORK

The event-driven online VFL framework is designed, and the procedures for both clients and servers are formalized in the algorithm 1.[3] When an event occurs at round $t$, the activated client $m$ will receive the data $x_m^t$. Subsequently, it sends the embedding $h_m(w_m^t; x_m^t)$ to the server. The server then requests embeddings from the passive clients $m \in \bar{A}_t \subset [M]$. After gathering the embedding,

---

[3]A synchronous version of algorithm 1 is provided in the Appendix C.1.

the server calculates the partial derivative of $f^t(\cdot)$ w.r.t. its local model $w_0^t$ and w.r.t. the output of the activated clients $h_m(w_m^t; x_m^t)$. Subsequently, the server updates the buffer for the sequence of partial derivatives in DLR. Following this, the server updates its local model with the partial derivative w.r.t. its model $w_0$ (Eq. 4) and sends the partial derivative w.r.t. the client's output $v_m^t$ to the activated client. After receiving from the server, each activated client $m \in A_t$ calculates the partial derivative w.r.t. their parameter via chain rule (Eq. 5), and then they update their parameter accordingly. At each round, we denote the set of the activated client as $A_t \subset [M]$, where $[M]$ represents the set of all clients. The set of passive clients is denoted by $\bar{A}_t = [M] \setminus A_t$.

---

**Algorithm 1** Event-driven online VFL

---

**Input:** window length $l$, coefficient $\alpha$, learning rate $\{\eta_m\}_{m=0}^M$,
**Output:** server model $w_0$, client models $w_m \in [M]$
 0: initialize model $w_m$ for all participants $m \in \{0, 1, ...M\}$
 1: **Client procedure:**
 2:   **if** activated by an event **then:**
 3:     sampling the environment to obtain $x_m^t$
 4:     send $h_m(w_m^t; x_m^t)$ to the server
 5:     receive $v_m^t$ from the server
 6:     enqueue $v_m^t \cdot \frac{\partial h_m(w_m^t; x_m^t)}{\partial w_m^t}$ into the client's buffer
 7:     calculate $\nabla_{w_m} \tilde{S}_{t,l,\alpha}(w_0^t, \mathbf{w}^t)$ with client's buffer
 8:     update the parameter $w_m^{t+1} \leftarrow w_m^t - \eta_m \cdot \nabla_{w_m} S_{t,l,\alpha}(w_0^t, \mathbf{w}^t)$
 9:   **else if** the server's query $t$ is received **then:**
10:     sampling the environment to obtain $x_m^t$
11:     send $h_m(w_m^t; x_m^t)$ to the server
12:     enqueue $\mathbf{0}$ to the client's buffer
13: **Server procedure:**
14:   **when** server receives $h_m(w_m^t; x_m^t)$ from activated client $m \in A_t$, **do:**
15:     send query $t$ to all passive client $m \in \bar{A}_t$
16:     calculate $\nabla_{w_0} f^t(w_0^{t-i}, \mathbf{w}^{t-i})$
17:     updates its model $w_0^{t+1} \leftarrow w_0^t - \eta_0 \cdot \nabla_{w_0} S_{t,l,\alpha}(w_0^t, \mathbf{w}^t)$
18:     sends $v_m = \frac{\partial S_{t,l,\alpha}(w_0^t, \tilde{\mathbf{w}}^t)}{\partial h_m(w_m^t; x_m^t)}$ to all activated clients $m \in A_t$

---

## 4 REGRET ANALYSIS

### 4.1 ASSUMPTION

Assumption 1 to 5 are the assumptions for analysis of the dynamic local regret bound under the non-convex case. Specifically, Assumption 1 is used for modeling the smoothness of the loss function $f(\cdot)$, with which we can link the difference of the gradients with the difference of the input in the definition domain. These are the basic assumptions for solving the non-convex optimization problem in VFL (Liu et al., 2019; Zhang et al., 2021a; Chen et al., 2020; Castiglia et al., 2022). Assumption 2 and assumption 3 are the common assumptions in the analysis of stochastic non-convex optimization (Aydore et al., 2019; Hazan et al., 2017). The unbiased gradient assumption means that the expected value of the stochastic gradient equals the true gradient for the underlying distribution of the sample. The bounded variance assumption ensures that the variability in the stochastic gradient estimates is limited. Assumption 4 is used for bounding the magnitude of the gradient for all participant's models. This is a common assumption for the non-convex optimization of VFL (Gu et al., 2020; Castiglia et al., 2022; Wang et al., 2023; Zhang et al., 2021a) and the regret analysis for online learning in VFL (Wang & Xu, 2023). This assumption is specifically employed to bound the difference between the gradient with missing elements (due to the event-driven framework) and the ideal gradient without such omissions. Assumption 5 assumes that the loss value is bounded, which is mainly used to bound the values of the difference between the sliding window average and to simplify the theoretical result. This is a common assumption in the regret analysis in online learning under non-convex case (Aydore et al., 2019; Hazan et al., 2017).

**Assumption 1.** *Lipschitz Gradient:* $\nabla f^t$ *is L-Lipschitz continuous w.r.t. all the parameter, i.e., there exists a constant L for* $\forall [w_0, \mathbf{w}], [w_0', \mathbf{w}']$ *such that*

$$\left\| \nabla_{[w_0, \mathbf{w}]} f^t(w_0, \mathbf{w}) - \nabla_{[w_0, \mathbf{w}]} f^t(w_0', \mathbf{w}') \right\| \leq L \left\| [w_0, \mathbf{w}] - [w_0', \mathbf{w}'] \right\| \tag{6}$$

**Assumption 2.** *Unbiased gradient: The stochastic gradient* $\hat{\nabla} f^t(w_0, \mathbf{w})$ *obtained at each iteration is an unbiased estimator of the true gradient of the function* $f^t(w_0, \mathbf{w})$, *i.e., for all* $[w_0, \mathbf{w}]$, *we have*

$$\mathbb{E}\left[ \hat{\nabla} f^t(w_0, \mathbf{w}) \right] = \nabla f^t(w_0, \mathbf{w}), \tag{7}$$

*where* $\nabla f^t(w_0, \mathbf{w})$ *is the true gradient of the global objective function.*

**Assumption 3.** *Bounded variance: The variance of the stochastic gradient* $\hat{\nabla} f^t(w_0, \mathbf{w})$ *obtained at each iteration is bounded, i.e., there exists a constant* $\sigma^2$ *such that for all* $[w_0, \mathbf{w}]$, *we have*

$$\mathbb{E}\left[ \left\| \hat{\nabla} f^t(w_0, \mathbf{w}) - \nabla f^t(w_0, \mathbf{w}) \right\|^2 \right] \leq \sigma^2. \tag{8}$$

*This ensures that the noise in the gradient estimation is controlled.*

**Assumption 4.** *Bounded gradient: The gradient of the objective function* $f^t(w_0, \mathbf{w})$ *is bounded, i.e. there exist positive constants* $\mathbf{G}$ *such that the following inequalities hold.*

$$\left\| \nabla_{[w_0, \mathbf{w}]} f^t(w_0, \mathbf{w}) \right\| \leq \mathbf{G} \tag{9}$$

**Assumption 5.** *Bounded loss: For all* $w_0 \in \mathbb{R}^{d_0}$ *and* $\mathbf{w} = [w_1, \cdots w_M]$, *where* $w_m \in \mathbb{R}^{d_m}$ *for* $m \in [M]$, *the loss* $f^t(w_0, \mathbf{w})$ *is bounded:*

$$|f^t(w_0, \mathbf{w})| < D \tag{10}$$

### 4.2 THEOREM

**Theorem 1.** *Dynamic local regret bound: Under Assumption 1 - Assumption 5, solving the event-driven online vertical federated learning problem with Algorithm 1. Select constant* $\eta_t = \eta$, *define* $p_{\min} = \min p_m$, $p_{\max} = \max p_m$, *let* $\alpha \to 1^-$.

$$DLR_l(T) \leq \frac{T}{W p_{\min}} \left( \frac{8D}{\eta_t} + 2L\eta_t p_{\max} \sigma^2 \right) + 2TL\eta_t \frac{p_{\max}}{p_{\min}} \mathbf{G}^2 \tag{11}$$

*Remark* 1. The last term in Eq. 11 arises from bounding the error between the gradient with missing elements in the event-driven framework and the ideal gradient without missing elements.

**Corollary 1.** *Select* $l = T^{\frac{1}{2}}$ *and* $\eta_t = T^{-\frac{1}{4}}$. *Let* $\alpha \to 1^-$.

$$DLR_l(T) = \mathcal{O}(T^{\frac{3}{4}}) \tag{12}$$

*Remark* 2. Corollary 1 suggests a sub-linear growth rate compared to the linear regret bound $\mathcal{O}(T)$, which demonstrates an effective regret minimization.

The proof of the Theorem 1 is in Appendix A. For the completeness of our analysis, we further provide the prevalent regret analysis of the event-driven online VFL in the convex case in Appendix B.

## 5 EXPERIMENT

### 5.1 EXPERIMENT SETUP

Supplementary experimental details, including the algorithms of the baselines specifically adapted for VFL, are provided in Appendix C. Furthermore, supplementary experiments addressing secondary aspects, such as ablation studies on the DLR parameter $(l, \alpha)$ and experiments on other datasets (SUSY, HIGGS), can be found in Appendix D.

**Dataset** We leveraged the Infinite MNIST (i-MNIST) dataset (Loosli et al., 2007) to assess the performance of the proposed methodologies in the context of VFL. The i-MNIST dataset extends the MNIST dataset by providing an endless stream of handwritten digit images along with their corresponding labels. To convert the i-MNIST dataset into a distributed dataset, each image was first flattened into a one-dimensional vector. This vector was then divided into four equal segments to ensure an even distribution of features across the clients. Each client was assigned one of the four feature partitions, while the server was assigned the label. Experiments on other practical online learning datasets, including the SUSY and HIGGS datasets, are provided in Appendix D.2.

**Model architecture** In our online VFL framework, there were four clients and one server. On the client side, the models consisted of a one-layer perceptron that took flattened features as input and generated 64-dimensional embeddings using ReLU activation. On the server side, a two-layer multi-layer perceptron was employed. The first layer took the concatenated client embeddings as input, produced an output of 256 units, and applied ReLU activation. The subsequent layer generated class logits using softmax activation. The loss function used by the server was cross-entropy.

**Baselines** The baselines comprised three different online VFL frameworks and three add-ons to those frameworks concerning the activation of participants, resulting in a total of nine baselines. The first online VFL framework applied online gradient descent in VFL (referred to as "OGD" for brevity). The second online VFL framework adapted Static Local Regret (Hazan et al., 2017) to online VFL (referred to as "SLR"). The third online VFL framework was our VFL framework incorporating dynamic local regret (referred to as "DLR").

The three activation add-ons included "Full" activation, "Random" activation, and "Event" activation. The "Full" activation represented the most common scenario where all clients were activated and received the synchronous data stream. The "Random" activation sets the activation probability for all clients to a constant value, i.e. $p_m = p$, which directly interpreted the theoretical result from theorem 1. The "Event" activation was an activation mechanism that we designed to simulate the activation in the sensor network. Many sensors are characterized by activation in response to detecting peaks (Suh, 2007) or stimuli (Heemels et al., 2012). Inspired by this, we implemented the "Event" activation in which a client is activated when the average of its input features exceeds a threshold, denoted as $\Gamma$, which mimics the stimuli from the surroundings.

To be more specific, the OGD-Full[4] framework was adapted from (Wang & Xu, 2023), customized to suit our general VFL setting. The OGD-Random and OGD-Event[5] were obtained by incorporating partial activation into the OGD-based online VFL. We also designed the SLR[6] baselines by incorporating Local Regret (Hazan et al., 2017) into the VFL with different activation schemes. DLR-Full was the model obtained by directly adapting the DLR to the synchronous VFL framework. DLR-Random and DLR-Event were the main contributions of our work, which incorporate partial client activation ("Random" or "Event") into VFL.

**Training procedure** We followed the standard online learning setting, wherein at each round $t$, client $m$ received only the corresponding feature of a single sample, rather than a batch. Each trial comprised a total of 2,000,000 non-repetitive samples. The learning rate $\eta$ was tuned from $\{1, 0.1, 0.01, 0.001, \ldots\}$. The length of the exponential weighted sliding window for the DLR was tuned from $\{10, 50, 100, 150\}$. The activation probability $p$ for the "Random" activation was selected from $\{0.25, 0.5, 0.75\}$. The activation threshold $\Gamma$ was tuned from $\{-0.2, 0, 0.2, 0.4, 0.6, 0.8\}$.

We conducted experiments on both a stationary data stream and a synthetic non-stationary data stream. In section 5.2. we used the original i-MNIST dataset as the stationary data stream, where each class had an equal probability of being sampled ($\frac{1}{10}$) at each round. In section 5.3 we synthesized a non-stationary data stream by altering the class sampling probabilities every 50 rounds. At each stage, the sampling probability for each class was drawn from a uniform distribution $U(0, 1)$ and then normalized. Data was then sampled based on the updated probabilities.

**Metrics** To evaluate the algorithm's performance, we employed the run-time error rate (prequential) during online learning. This error rate was averaged and reported every 20,000 samples, providing insights into performance at each stage of the training. Additionally, the accumulated error rate, representing the overall error for the entire training process, served as a metric for evaluating overall performance. In terms of computational efficiency, we reported the total computation time for all clients, including both active and passive clients at each round. In terms of communication efficiency, we recorded the total communication cost for the VFL framework throughout the entire training process, specifically measuring the size of all communication messages exchanged between the server and the clients.

---

[4] Algorithm for OGD-Full are provided in Algorithm 4 in Appendix C.2.

[5] Algorithm for OGD-Event are provided in Algorithm 2 in Appendix B.

[6] Algorithm for SLR-Event are provided in Algorithm 5 in Appendix C.3.

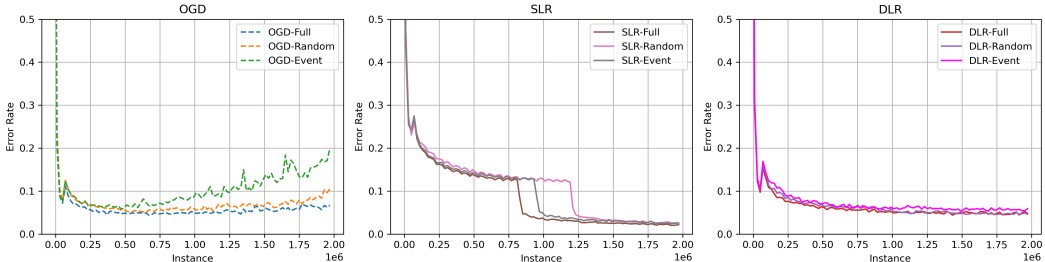

Figure 2: Run-time error rate under stationary data stream

Table 1: Performance metrics for online VFL under stationary data stream

| | OGD | | | SLR | | | DLR | | |
| Metric | Full | Random | Event | Full | Random | Event | Full | Random | Event |
|---|---|---|---|---|---|---|---|---|---|
| Accum. Error Rate | 0.0575 | 0.0696 | 0.1027 | 0.0846 | 0.1083 | 0.0945 | 0.0618 | **0.0649** | **0.0718** |
| Client Comp. (s) | 4339.20 | 2652.38 | 3182.46 | 4565.42 | 2682.47 | 2992.73 | 4565.42 | 2682.47 | 2992.73 |
| Client Comm. (MB) | 1953.13 | **1465.04** | **1686.86** | 78125.02 | 58599.36 | 61565.37 | 1953.13 | **1464.73** | **1539.13** |

## 5.2 RESULT ON STATIONARY DATA STREAM

Figure 2 illustrates a comparison of the run-time error rates across all frameworks within a stationary data stream scenario. The x-axis shows the number of observed instances, while the y-axis represents the corresponding run-time error rates. From the analysis of the results, it is evident that in the "Full" activation framework, all models demonstrate stable convergence behavior. However, under the partial activation scheme, both SLR and DLR show superior convergence curves compared to OGD. Notably, DLR converges more rapidly than SLR, entering the convergence phase earlier.

Table 1 presents the performance metrics for the entire training process, including the accumulated error rate, the total computational time for the client, and the total communication cost for the entire VFL framework. The accumulated error further illustrates that the DLR exhibits greater stability under the partial activation scheme. Specifically, the accumulated error rate for the DLR remains approximately $0.06$ when employing partial activation, whereas the accumulated error rate for OGD undergoes a significant reduction from $0.0575$ to $0.1027$ when implementing the partial activation scheme. By comparing DLR and SLR, we observed that DLR converges more quickly, resulting in a lower average error rate. Specifically, due to the buffer design, SLR required the communication of the entire buffer between the server and the client in each round[7]. Consequently, the communication volume of SLR was an order of magnitude higher than that of DLR and OGD. Compared to "Full" activation, the partial activation approach results in a slightly higher error rate; however, it significantly reduces client computation and overall communication costs.

## 5.3 RESULT ON NON-STATIONARY DATA STREAM

Figure 3 compares the run-time error rates across all the baselines under the non-stationary data stream case. The DLR and SLR frameworks generally outperformed OGD, demonstrating greater stability under the partial activation scheme for clients. Towards the end of the training process, OGD became increasingly unstable, particularly when partial activation add-ons were used. In contrast, both DLR and SLR maintained stability across all activation schemes. Although SLR was able to converge in non-stationary environments, it was less adaptable to dynamic environments compared to DLR, as its convergence was slower at the beginning of training.

Table 2 presents the performance metrics of the frameworks for comparison. Overall, the DLR method demonstrated a lower accumulated error rate compared to OGD and SLR, primarily due to its stability against non-stationary data streams and partial activation. The communication cost of SLR was significantly higher than that of OGD and DLR, consistent with previous experimental results. When comparing full activation with the partial activation scheme, it was observed that

---

[7]The step that incurs a high communication cost in SLR is highlighted in Algorithm 5 in Appendix C.3.

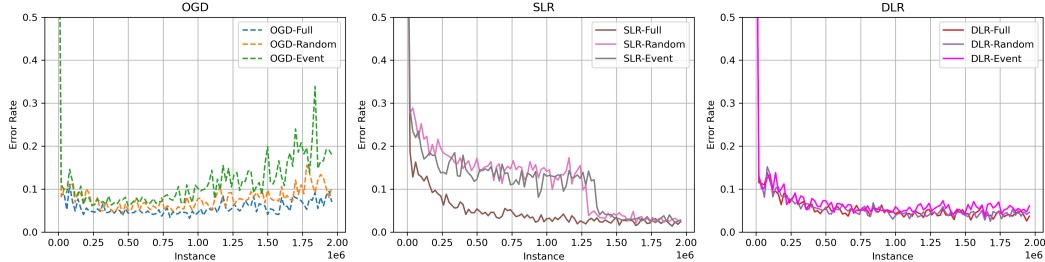

Figure 3: Run-time error rate under non-stationary data stream

Table 2: Performance metrics for online VFL under non-stationary data stream

| Metric | OGD | | | SLR | | | DLR | | |
|---|---|---|---|---|---|---|---|---|---|
| | Full | Random | Event | Full | Random | Event | Full | Random | Event |
| Accum. Error Rate | 0.0592 | 0.0788 | 0.1191 | 0.0481 | 0.1159 | 0.1118 | 0.0553 | **0.0561** | **0.0681** |
| Client Comp. (s) | 4406.14 | 2822.29 | 3168.34 | 3497.49 | 2172.44 | 2103.50 | 4558.20 | 2872.79 | 2698.96 |
| Total Comm. (MB) | 1953.12 | **1464.71** | **1543.32** | 78124.61 | 58589.34 | 61748.22 | 1953.12 | **1465.02** | **1413.52** |

the partial activation approach typically resulted in a slightly higher total error rate during training. However, it significantly reduced both computational and communication costs, making it a more practical and efficient solution for real-world applications.

## 5.4 ENHANCING COMPUTATION-COMMUNICATION EFFICIENCY WITH PARTIAL ACTIVATION

In the partial activation approach ("Random" and "Event"), fewer clients participated in the training of each epoch, therefore reducing both computation and communication costs compared to the framework with "Full" activation.

**Activation probability** $p$   In the "Random" activation framework, the activation probability $p$ determines the likelihood of a client being activated in each round. A higher value of $p$ increases the probability of activation, while a lower value decreases it. We conducted the study on the activation probability within the DLR-Random Framework under the non-stationary data stream, using a window length of $l = 10$ and an attenuation coefficient $\alpha = 0.95$. We selected $p$ from the range $\{0.25, 0.50, 0.75, 1.00\}$.

This observation indicates that the run-time error rate among the DLR-Random models remains consistent across varying activation probabilities. Table 3 presents the corresponding performance metrics of those trials. This indicates that a small activation probability proportionally decreases the computational cost for the client, albeit with a slight increase in the accumulated error rate. Additionally, both client computation time and communication decrease, as passive clients do not participate in the backward process.

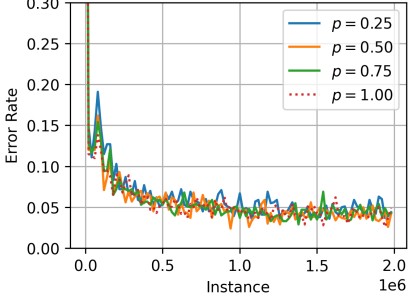

Figure 4: Activation probability $p$

Table 3: Performance metrics for different activation probability

| Activation Probability | Accum. Error Rate | Client Comp. (s) | Total Comm. (MB) |
|---|---|---|---|
| 0.25 | 0.0622 | 1903.91 | 1220.75 |
| 0.50 | 0.0561 | 2872.79 | 1465.02 |
| 0.75 | 0.0560 | 3788.07 | 1709.04 |
| 1.00 | 0.0553 | 4558.20 | 1953.12 |

**Event activation threshold** $\Gamma$ Using the DLR-Event framework with a window length of $l = 10$ and an attenuation coefficient of $\alpha = 0.95$, we examined the impact of varying the activation threshold. Table 4 presents the performance metrics for the DLR-Event with different $\Gamma$. As the threshold increased, fewer clients were activated per round, leading to an overall increase in the framework's accumulated error rate while reducing its computational cost and communication costs. We further provided an examination of the activation rate for each client across various activation thresholds. As the threshold surpasses $0.6$, few clients were activated in each round, the framework will mostly rely on the server for learning, with clients primarily offering a nearly invariant mapping of the input features. Conversely, as the threshold decreases, clients can be activated more frequently.

Table 4: Performance metrics for activation threshold

| Threshold $\Gamma$ | Accum. Error Rate | Client Comp. (s) | Total Comm. (MB) |
|---|---|---|---|
| 0.6 | 0.1057 | 928.67 | 1017.76 |
| 0.2 | 0.0738 | 1852.15 | 1211.55 |
| $-0.2$ | 0.0595 | 3752.07 | 1689.42 |

## 6 LIMITATIONS

Due to the inherent nature of VFL, wherein the clients possess non-intersecting feature sets, the involvement of passive clients in each round remains unavoidable. Omitting passive participants leads to missing input at the server's model, which is known as the "incomplete view" problem in VFL or the "non-overlapping sample" problem within the offline VFL setting. Although some research on VFL has attempted to address the incomplete view problem through knowledge distillation (KD) (Ren et al., 2022), self-supervised learning (SSL) (He et al., 2022; Li et al., 2022; Kang et al., 2022) and semi-supervised learning (Li et al., 2024), these approaches entail more complex models and higher computation-communication costs. For instance, Ren et al. (2022) train student models for each case with an incomplete view via KD, which imposes a significant computational cost, especially when the number of clients is large. Another example is the work by He et al. (2022), which implements SSL within the VFL framework. This approach requires communication between participants during the computation-intensive pretraining stage. Consequently, in scenarios where participants have limited computational resources, there is no perfect solution to this problem.

The future direction of event-driven online VFL focuses on examining the complete independent data streams for each client by addressing the incomplete view problem within the VFL framework. Resolving this problem will eliminate the need for passive client participation in each event, thereby reducing computation and communication costs and enabling the implementation of fully asynchronous online VFL.

## 7 CONCLUSION

Adapting online learning to vertical federated learning poses challenges due to the unique nature of VFL. The clients may not receive data streaming synchronously, as data generated by an "event" typically pertains to only a subset of the clients. To address these challenges, we proposed the Event-Driven Online Vertical Federated Learning framework. Within this framework, only a subset of clients is activated by the event in each round, with others passively participating in the learning process. Moreover, we incorporate dynamic local regret to address non-convex models in a non-stationary environment, which further enhances the adaptability of our framework for real-world applications. Through comprehensive regret analysis, we have derived the regret bound of $\mathcal{O}(T^{\frac{3}{4}})$ for dynamic local regret under the non-convex case. Through extensive experiments, we have demonstrated the effectiveness of our approach in reducing communication-computation costs and achieving strong performance in non-stationary environments. Overall, our contributions pave the way for more practical and adaptable implementations of VFL in real-world scenarios.

ACKNOWLEDGMENTS

This work has been supported by the Natural Sciences and Engineering Research Council of Canada (NSERC), Discovery Grants program.

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

## A    REGRET ANALYSIS: NON-CONVEX CASE WITH DYNAMIC LOCAL REGRET

*Proof of Theorem 1:*

Taking Expectation w.r.t $x_t$.

$$\mathbb{E}_{x_t}\left[S_{t,l,\alpha}(w_0^{t+1}, \mathbf{w}^{t+1}) - S_{t,l,\alpha}(w_0^t, \mathbf{w}^t)\right]$$

$$\overset{1)}{\leq} \mathbb{E}_{x_t}\left\langle \nabla S_{t,l,\alpha}(w_0^t, \mathbf{w}^t), [w_0^{t+1}, \mathbf{w}^{t+1}] - [w_0^t, \mathbf{w}^t]\right\rangle + \frac{L}{2}\mathbb{E}_{x_t}\left\|[w_0^{t+1}, \mathbf{w}^{t+1}] - [w_0^t, \mathbf{w}^t]\right\|^2$$

$$\overset{2)}{=} -\eta_t \mathbb{E}_{x_t}\left\langle \nabla S_{t,l,\alpha}(w_0^t, \mathbf{w}^t), \hat{\nabla}_{w_0} S_{t,l,\alpha}(w_0^t, \mathbf{w}^t)\right\rangle - \eta_t \mathbb{E}_{x_t}\left\langle \nabla S_{t,l,\alpha}(w_0^t, \mathbf{w}^t), \sum_{m \in A_t} \hat{\nabla}_{w_m} \tilde{S}_{t,l,\alpha}(w_0^t, \mathbf{w}^t)\right\rangle$$

$$+ \frac{L\eta_t^2}{2}\mathbb{E}_{x_t}\left\|\hat{\nabla}_{w_0} S_{t,l,\alpha}(w_0^t, \mathbf{w}^t)\right\|^2 + \frac{L\eta_t^2}{2}\sum_{m \in A_t}\mathbb{E}_{x_t}\left\|\hat{\nabla}_{w_m} \tilde{S}_{t,l,\alpha}(w_0^t, \mathbf{w}^t)\right\|^2$$

$$= \underbrace{-\eta_t\mathbb{E}_{x_t}\left\langle \nabla S_{t,l,\alpha}(w_0^t, \mathbf{w}^t), \hat{\nabla}_{w_0} S_{t,l,\alpha}(w_0^t, \mathbf{w}^t)\right\rangle + \frac{L\eta_t^2}{2}\mathbb{E}_{x_t}\left\|\hat{\nabla}_{w_0} S_{t,l,\alpha}(w_0^t, \mathbf{w}^t)\right\|^2}_{a)}$$

$$\underbrace{-\eta_t\mathbb{E}_{x_t}\left\langle \nabla S_{t,l,\alpha}(w_0^t, \mathbf{w}^t), \sum_{m \in A_t} \hat{\nabla}_{w_m} \tilde{S}_{t,l,\alpha}(w_0^t, \mathbf{w}^t)\right\rangle + \frac{L\eta_t^2}{2}\sum_{m \in A_t}\mathbb{E}_{x_t}\left\|\hat{\nabla}_{w_m} \tilde{S}_{t,l,\alpha}(w_0^t, \mathbf{w}^t)\right\|^2}_{b)}$$

$$\tag{13}$$

where 1) applies the Lipschitz Continuous of $S_{t,l,\alpha}(w_0^t, \mathbf{w}^t)$,
2) applies the update in one round of the event-driven online VFL.

For a)

$$-\eta_t\mathbb{E}_{x_t}\left\langle \nabla S_{t,l,\alpha}(w_0^t, \mathbf{w}^t), \hat{\nabla}_{w_0} S_{t,l,\alpha}(w_0^t, \mathbf{w}^t)\right\rangle + \frac{L\eta_t^2}{2}\mathbb{E}_{x_t}\left\|\hat{\nabla}_{w_0} S_{t,l,\alpha}(w_0^t, \mathbf{w}^t)\right\|^2$$

$$\overset{1)}{=} -\eta_t\mathbb{E}_{x_t}\left\|\nabla S_{t,l,\alpha}(w_0^t, \mathbf{w}^t)\right\|^2$$

$$+ \frac{L\eta_t^2}{2}\mathbb{E}_{x_t}\left\|\hat{\nabla}_{w_0} S_{t,l,\alpha}(w_0^t, \mathbf{w}^t) - \nabla_{w_0} S_{t,l,\alpha}(w_0^t, \mathbf{w}^t) + \nabla_{w_0} S_{t,l,\alpha}(w_0^t, \mathbf{w}^t)\right\|^2$$

$$\overset{2)}{\leq} -\eta_t\mathbb{E}_{x_t}\left\|\nabla S_{t,l,\alpha}(w_0^t, \mathbf{w}^t)\right\|^2$$

$$+ L\eta_t^2\mathbb{E}_{x_t}\left\|\hat{\nabla}_{w_0} S_{t,l,\alpha}(w_0^t, \mathbf{w}^t) - \nabla_{w_0} S_{t,l,\alpha}(w_0^t, \mathbf{w}^t)\right\|^2 + L\eta_t^2\mathbb{E}_{x_t}\left\|\nabla_{w_0} S_{t,l,\alpha}(w_0^t, \mathbf{w}^t)\right\|^2$$

$$= -\left(\eta_t - L\eta_t^2\right)\mathbb{E}_{x_t}\left\|\nabla S_{t,l,\alpha}(w_0^t, \mathbf{w}^t)\right\|^2$$

$$+ L\eta_t^2\mathbb{E}_{x_t}\left\|\hat{\nabla}_{w_0} S_{t,l,\alpha}(w_0^t, \mathbf{w}^t) - \nabla_{w_0} S_{t,l,\alpha}(w_0^t, \mathbf{w}^t)\right\|^2$$

$$\tag{14}$$

where 1) applies assumption 2 (unbiased gradient),
2) applies $\|a + b\|^2 \leq 2\|a\|^2 + 2\|b\|^2$.

For b)

$$-\eta_t\mathbb{E}_{x_t}\left\langle \nabla S_{t,l,\alpha}(w_0^t, \mathbf{w}^t), \sum_{m \in A_t} \hat{\nabla}_{w_m} \tilde{S}_{t,l,\alpha}(w_0^t, \mathbf{w}^t)\right\rangle + \frac{L\eta_t^2}{2}\sum_{m \in A_t}\mathbb{E}_{x_t}\left\|\hat{\nabla}_{w_m} \tilde{S}_{t,l,\alpha}(w_0^t, \mathbf{w}^t)\right\|^2$$

$$= -\eta_t\mathbb{E}_{x_t}\left\langle \nabla S_{t,l,\alpha}(w_0^t, \mathbf{w}^t), \sum_{m \in A_t} \hat{\nabla}_{w_m} \tilde{S}_{t,l,\alpha}(w_0^t, \mathbf{w}^t) - \sum_{m \in A_t} \hat{\nabla}_{w_m} S_{t,l,\alpha}(w_0^t, \mathbf{w}^t) + \sum_{m \in A_t} \hat{\nabla}_{w_m} S_{t,l,\alpha}(w_0^t, \mathbf{w}^t)\right\rangle$$

$$+ \frac{L\eta_t^2}{2}\sum_{m \in A_t}\mathbb{E}_{x_t}\left\|\hat{\nabla}_{w_m} \tilde{S}_{t,l,\alpha}(w_0^t, \mathbf{w}^t) - \hat{\nabla}_{w_m} S_{t,l,\alpha}(w_0^t, \mathbf{w}^t) + \hat{\nabla}_{w_m} S_{t,l,\alpha}(w_0^t, \mathbf{w}^t)\right\|^2$$

$$
\overset{1)}{=} - \eta_t \mathbb{E}_{x_t} \left\| \sum_{m \in A_t} \nabla_{w_m} S_{t,l,\alpha}(w_0^t, \mathbf{w}^t) \right\|^2
$$

$$
- \eta_t \mathbb{E}_{x_t} \left\langle \nabla S_{t,l,\alpha}(w_0^t, \mathbf{w}^t), \sum_{m \in A_t} \hat{\nabla}_{w_m} \tilde{S}_{t,l,\alpha}(w_0^t, \mathbf{w}^t) - \sum_{m \in A_t} \hat{\nabla}_{w_m} S_{t,l,\alpha}(w_0^t, \mathbf{w}^t) \right\rangle
$$

$$
+ \frac{L \eta_t^2}{2} \sum_{m \in A_t} \mathbb{E}_{x_t} \left\| \hat{\nabla}_{w_m} \tilde{S}_{t,l,\alpha}(w_0^t, \mathbf{w}^t) - \hat{\nabla}_{w_m} S_{t,l,\alpha}(w_0^t, \mathbf{w}^t) + \hat{\nabla}_{w_m} S_{t,l,\alpha}(w_0^t, \mathbf{w}^t) \right\|^2
$$

$$
\overset{2)}{=} - \eta_t \mathbb{E}_{x_t} \left\| \sum_{m \in A_t} \nabla_{w_m} S_{t,l,\alpha}(w_0^t, \mathbf{w}^t) \right\|^2
$$

$$
- \eta_t \mathbb{E}_{x_t} \left\langle \nabla S_{t,l,\alpha}(w_0^t, \mathbf{w}^t), \sum_{m \in A_t} \hat{\nabla}_{w_m} \tilde{S}_{t,l,\alpha}(w_0^t, \mathbf{w}^t) - \sum_{m \in A_t} \hat{\nabla}_{w_m} S_{t,l,\alpha}(w_0^t, \mathbf{w}^t) \right\rangle
$$

$$
+ L \eta_t^2 \sum_{m \in A_t} \mathbb{E}_{x_t} \left\| \hat{\nabla}_{w_m} \tilde{S}_{t,l,\alpha}(w_0^t, \mathbf{w}^t) - \hat{\nabla}_{w_m} S_{t,l,\alpha}(w_0^t, \mathbf{w}^t) \right\|^2
$$

$$
+ L \eta_t^2 \sum_{m \in A_t} \mathbb{E}_{x_t} \left\| \hat{\nabla}_{w_m} S_{t,l,\alpha}(w_0^t, \mathbf{w}^t) \right\|^2
$$

$$
\overset{3)}{=} - \eta_t \mathbb{E}_{x_t} \left\| \sum_{m \in A_t} \nabla_{w_m} S_{t,l,\alpha}(w_0^t, \mathbf{w}^t) \right\|^2
$$

$$
- \eta_t \mathbb{E}_{x_t} \left\langle \nabla S_{t,l,\alpha}(w_0^t, \mathbf{w}^t), \sum_{m \in A_t} \hat{\nabla}_{w_m} \tilde{S}_{t,l,\alpha}(w_0^t, \mathbf{w}^t) - \sum_{m \in A_t} \hat{\nabla}_{w_m} S_{t,l,\alpha}(w_0^t, \mathbf{w}^t) \right\rangle
$$

$$
+ L \eta_t^2 \sum_{m \in A_t} \mathbb{E}_{x_t} \left\| \hat{\nabla}_{w_m} \tilde{S}_{t,l,\alpha}(w_0^t, \mathbf{w}^t) - \hat{\nabla}_{w_m} S_{t,l,\alpha}(w_0^t, \mathbf{w}^t) \right\|^2
$$

$$
+ L \eta_t^2 \sum_{m \in A_t} \mathbb{E}_{x_t} \left\| \nabla_{w_m} S_{t,l,\alpha}(w_0^t, \mathbf{w}^t) \right\|^2 + L \eta_t^2 \sum_{m \in A_t} \mathbb{E}_{x_t} \left\| \hat{\nabla}_{w_m} S_{t,l,\alpha}(w_0^t, \mathbf{w}^t) - \nabla_{w_m} S_{t,l,\alpha}(w_0^t, \mathbf{w}^t) \right\|^2
$$

$$
= - \left( \eta_t - L \eta_t^2 \right) \mathbb{E}_{x_t} \left\| \sum_{m \in A_t} \nabla_{w_m} S_{t,l,\alpha}(w_0^t, \mathbf{w}^t) \right\|^2
$$

$$
- \eta_t \mathbb{E}_{x_t} \left\langle \nabla S_{t,l,\alpha}(w_0^t, \mathbf{w}^t), \sum_{m \in A_t} \hat{\nabla}_{w_m} \tilde{S}_{t,l,\alpha}(w_0^t, \mathbf{w}^t) - \sum_{m \in A_t} \hat{\nabla}_{w_m} S_{t,l,\alpha}(w_0^t, \mathbf{w}^t) \right\rangle
$$

$$
+ L \eta_t^2 \sum_{m \in A_t} \mathbb{E}_{x_t} \left\| \hat{\nabla}_{w_m} \tilde{S}_{t,l,\alpha}(w_0^t, \mathbf{w}^t) - \hat{\nabla}_{w_m} S_{t,l,\alpha}(w_0^t, \mathbf{w}^t) \right\|^2
$$

$$
+ L \eta_t^2 \sum_{m \in A_t} \mathbb{E}_{x_t} \left\| \hat{\nabla}_{w_m} S_{t,l,\alpha}(w_0^t, \mathbf{w}^t) - \nabla_{w_m} S_{t,l,\alpha}(w_0^t, \mathbf{w}^t) \right\|^2
$$

$$
\tag{15}
$$

where 1) applies assumption 2 (unbiased gradient),
2) applies $\|a + b\|^2 \leq 2 \|a\|^2 + 2 \|b\|^2$,
3) $\mathbb{E}(X^2) = \mathbb{E}(X)^2 + \mathrm{Var}(X)$.

Plug in a) and b) and taking expectation w.r.t. the activated client $m \in A_t$, where the activation probability of client $m$ is $p_m$, we derive:

$$
\mathbb{E}_m \mathbb{E}_{x_t} \left[ S_{t,l,\alpha}(w_0^{t+1}, \mathbf{w}^{t+1}) - S_{t,l,\alpha}(w_0^t, \mathbf{w}^t) \right]
$$

$$
\leq - \left( \eta_t - L \eta_t^2 \right) \mathbb{E}_{x_t} \left\| \nabla_{w_0} S_{t,l,\alpha}(w_0^t, \mathbf{w}^t) \right\|^2
$$

$$+ L\eta_t^2 \mathbb{E}_{x_t} \left\| \hat{\nabla}_{w_0} S_{t,l,\alpha}(w_0^t, \mathbf{w}^t) - \nabla_{w_0} S_{t,l,\alpha}(w_0^t, \mathbf{w}^t) \right\|^2$$

$$- \left( \eta_t - L\eta_t^2 \right) \sum_{m \in [M]} p_m \mathbb{E}_{x_t} \left\| \nabla_{w_m} S_{t,l,\alpha}(w_0^t, \mathbf{w}^t) \right\|^2$$

$$- \eta_t \mathbb{E}_{x_t} \sum_{m \in [M]} p_m \left\langle \nabla_{w_m} S_{t,l,\alpha}(w_0^t, \mathbf{w}^t), \hat{\nabla}_{w_m} \tilde{S}_{t,l,\alpha}(w_0^t, \mathbf{w}^t) - \hat{\nabla}_{w_m} S_{t,l,\alpha}(w_0^t, \mathbf{w}^t) \right\rangle$$

$$+ L\eta_t^2 \sum_{m \in [M]} p_m \mathbb{E}_{x_t} \left\| \hat{\nabla}_{w_m} \tilde{S}_{t,l,\alpha}(w_0^t, \mathbf{w}^t) - \hat{\nabla}_{w_m} S_{t,l,\alpha}(w_0^t, \mathbf{w}^t) \right\|^2$$

$$+ L\eta_t^2 \sum_{m \in [M]} p_m \mathbb{E}_{x_t} \left\| \hat{\nabla}_{w_m} S_{t,l,\alpha}(w_0^t, \mathbf{w}^t) - \nabla_{w_m} S_{t,l,\alpha}(w_0^t, \mathbf{w}^t) \right\|^2$$

$$\overset{1)}{\leq} - \left( \eta_t - L\eta_t^2 \right) p_{\min} \mathbb{E}_{x_t} \left\| \nabla S_{t,l,\alpha}(w_0^t, \mathbf{w}^t) \right\|^2$$

$$+ L\eta_t^2 p_{\max} \mathbb{E}_{x_t} \left\| \hat{\nabla} S_{t,l,\alpha}(w_0^t, \mathbf{w}^t) - \nabla S_{t,l,\alpha}(w_0^t, \mathbf{w}^t) \right\|^2$$

$$- \eta_t \mathbb{E}_{x_t} \sum_{m \in [M]} p_m \left\langle \nabla_{w_m} S_{t,l,\alpha}(w_0^t, \mathbf{w}^t), \hat{\nabla}_{w_m} \tilde{S}_{t,l,\alpha}(w_0^t, \mathbf{w}^t) - \hat{\nabla}_{w_m} S_{t,l,\alpha}(w_0^t, \mathbf{w}^t) \right\rangle$$

$$+ L\eta_t^2 \sum_{m \in [M]} p_m \mathbb{E}_{x_t} \left\| \hat{\nabla}_{w_m} \tilde{S}_{t,l,\alpha}(w_0^t, \mathbf{w}^t) - \hat{\nabla}_{w_m} S_{t,l,\alpha}(w_0^t, \mathbf{w}^t) \right\|^2$$

$$\overset{2)}{\leq} - \left( \eta_t - L\eta_t^2 \right) p_{\min} \mathbb{E}_{x_t} \left\| \nabla S_{t,l,\alpha}(w_0^t, \mathbf{w}^t) \right\|^2$$

$$+ L\eta_t^2 p_{\max} \cdot \frac{\sigma^2}{W^2} \cdot \frac{1 - \alpha^{2l}}{1 - \alpha^2}$$

$$- \eta_t \mathbb{E}_{x_t} \sum_{m \in [M]} p_m \left\langle \nabla_{w_m} S_{t,l,\alpha}(w_0^t, \mathbf{w}^t), \hat{\nabla}_{w_m} \tilde{S}_{t,l,\alpha}(w_0^t, \mathbf{w}^t) - \hat{\nabla}_{w_m} S_{t,l,\alpha}(w_0^t, \mathbf{w}^t) \right\rangle$$

$$\underbrace{+ L\eta_t^2 \sum_{m \in [M]} p_m \mathbb{E}_{x_t} \left\| \hat{\nabla}_{w_m} \tilde{S}_{t,l,\alpha}(w_0^t, \mathbf{w}^t) - \hat{\nabla}_{w_m} S_{t,l,\alpha}(w_0^t, \mathbf{w}^t) \right\|^2}_{c)}$$

$$\overset{3)}{\leq} - \left( \eta_t - L\eta_t^2 \right) p_{\min} \mathbb{E}_{x_t} \left\| \nabla S_{t,l,\alpha}(w_0^t, \mathbf{w}^t) \right\|^2$$

$$+ L\eta_t^2 p_{\max} \cdot \frac{\sigma^2}{W^2} \cdot \frac{1 - \alpha^{2l}}{1 - \alpha^2}$$

$$- \eta_t \mathbb{E}_{x_t} \sum_{m \in [M]} p_m \left\langle \nabla_{w_m} S_{t,l,\alpha}(w_0^t, \mathbf{w}^t), \hat{\nabla}_{w_m} \tilde{S}_{t,l,\alpha}(w_0^t, \mathbf{w}^t) - \hat{\nabla}_{w_m} S_{t,l,\alpha}(w_0^t, \mathbf{w}^t) \right\rangle$$

$$+ L\eta_t^2 p_{\max} \mathbf{G}^2$$

$$\tag{16}$$

where 1) note that $m$ is in different dimension,
2) applies assumption 3 (bounded variance), and $\alpha$ is a weighted average of $l$ independently, therefore $\mathbb{E}_{x_t} \left\| \hat{\nabla} S_{t,l,\alpha}(w_0^t, \mathbf{w}^t) - \nabla S_{t,l,\alpha}(w_0^t, \mathbf{w}^t) \right\|^2 \leq \frac{\sigma^2}{W^2} \cdot \frac{1 - \alpha^{2l}}{1 - \alpha^2}$,
3) plug in c).

For c)

$$\sum_{m \in [M]} p_m \mathbb{E}_{x_t} \left\| \hat{\nabla}_{w_m} \tilde{S}_{t,l,\alpha}(w_0^t, \mathbf{w}^t) - \hat{\nabla}_{w_m} S_{t,l,\alpha}(w_0^t, \mathbf{w}^t) \right\|^2$$

$$= \sum_{m \in [M]} p_m \mathbb{E}_{x_t} \left\| \frac{1}{W} \sum_{i=0}^{l-1} \alpha^i \nabla_{w_m} f^{t-i}(w_0^{t-i}, \mathbf{w}^{t-i}) \cdot (1 - \gamma_m^{t-i}) \right\|^2$$

$$
= \frac{1}{W^2} \sum_{m \in [M]} p_m \mathbb{E}_{x_t} \left\| \sum_{i=0}^{l-1} \alpha^i \hat{\nabla}_{w_m} f^{t-i}(w_0^{t-i}, \mathbf{w}^{t-i}) \cdot (1 - \gamma_m^{t-i}) \right\|^2
$$

$$
\overset{1)}{\leq} \frac{1}{W^2} \sum_{m \in [M]} p_m \left( \sum_{i=0}^{l-1} \alpha^i (1 - \gamma_m^{t-i}) \right)^2 \left\| \hat{\nabla}_{w_m} f^{t-i}(w_0^{t-i}, \mathbf{w}^{t-i}) \right\|^2
$$

$$
\leq \frac{1}{W^2} \sum_{m \in [M]} p_m \left( \sum_{i=0}^{l-1} \alpha^i \right)^2 \left\| \hat{\nabla}_{w_m} f^{t-i}(w_0^{t-i}, \mathbf{w}^{t-i}) \right\|^2
$$

$$
\overset{2)}{\leq} \sum_{m \in [M]} p_m \left\| \hat{\nabla}_{w_m} f^{t-i}(w_0^{t-i}, \mathbf{w}^{t-i}) \right\|^2
$$

$$
\overset{3)}{\leq} \sum_{m \in [M]} p_{\max} \left\| \hat{\nabla}_{w_m} f^{t-i}(w_0^{t-i}, \mathbf{w}^{t-i}) \right\|^2
$$

$$
\overset{4)}{\leq} p_{\max} \mathbf{G}^2
$$

$$(17)$$

where 1) by triangle inequality $\left\| \sum_{i=0}^{l-1} \alpha^i \nabla_{w_m} f^{t-i}(w_0^{t-i}, \mathbf{w}^{t-i}) \cdot (1 - \gamma_m^{t-i}) \right\| \leq \sum_{i=0}^{l-1} \left\| \alpha^i \nabla_{w_m} f^{t-i}(w_0^{t-i}, \mathbf{w}^{t-i}) \cdot (1 - \gamma_m^{t-i}) \right\| \leq \sum_{i=0}^{l-1} \alpha^i (1 - \gamma_m^{t-i}) \left\| \nabla_{w_m} f^{t-i}(w_0^{t-i}, \mathbf{w}^{t-i}) \right\|$,
2) $W = \sum_{i=0}^{l-1} \alpha^i$,
3) denote $\max_m \{p_m\} = p_{\max}$,
4) applies assumption 4 (bounded gradient).

Taking expectation w.r.t. time step $t \sim \text{Unif}([T])$, and use $\mathbb{E}$ to denote the expectation $\mathbb{E}_t \mathbb{E}_m \mathbb{E}_{x_t}$. We can eliminate $\left\langle \nabla_{w_m} S_{t,l,\alpha}(w_0^t, \mathbf{w}^t), \hat{\nabla}_{w_m} \tilde{S}_{t,l,\alpha}(w_0^t, \mathbf{w}^t) - \hat{\nabla}_{w_m} S_{t,l,\alpha}(w_0^t, \mathbf{w}^t) \right\rangle$, because $\mathbb{E}_t \nabla \tilde{S}_{t,l,\alpha}(w_0^t, \mathbf{w}^t) = \mathbb{E}_t \nabla S_{t,l,\alpha}(w_0^t, \mathbf{w}^t)$.

$$
\mathbb{E} \left[ S_{t,l,\alpha}(w_0^{t+1}, \mathbf{w}^{t+1}) - S_{t,l,\alpha}(w_0^t, \mathbf{w}^t) \right] \leq - \left( \eta_t - L\eta_t^2 \right) p_{\min} \mathbb{E} \left\| \nabla S_{t,l,\alpha}(w_0^t, \mathbf{w}^t) \right\|^2
$$
$$
+ L\eta_t^2 p_{\max} \cdot \frac{\sigma^2}{W^2} \cdot \frac{1 - \alpha^{2l}}{1 - \alpha^2} + L\eta_t^2 p_{\max} \mathbf{G}^2
$$

$$(18)$$

Rearrange the above equation we have:

$$
\left( \eta_t - L\eta_t^2 \right) p_{\min} \mathbb{E} \left\| \nabla S_{t,l,\alpha}(w_0^t, \mathbf{w}^t) \right\|^2
$$
$$
\leq \mathbb{E} \left[ S_{t,l,\alpha}(w_0^t, \mathbf{w}^t) - S_{t,l,\alpha}(w_0^{t+1}, \mathbf{w}^{t+1}) \right]
$$
$$
+ L\eta_t^2 p_{\max} \cdot \frac{\sigma^2}{W^2} \cdot \frac{1 - \alpha^{2l}}{1 - \alpha^2} + L\eta_t^2 p_{\max} \mathbf{G}^2
$$
$$
\leq \underbrace{\mathbb{E} S_{t,l,\alpha}(w_0^t, \mathbf{w}^t) - \mathbb{E} S_{t+1,l,\alpha}(w_0^{t+1}, \mathbf{w}^{t+1})}_{d)} + \underbrace{\mathbb{E} S_{t+1,l,\alpha}(w_0^{t+1}, \mathbf{w}^{t+1}) - \mathbb{E} S_{t,l,\alpha}(w_0^{t+1}, \mathbf{w}^{t+1})}_{e)}
$$
$$
+ L\eta_t^2 p_{\max} \cdot \frac{\sigma^2}{W^2} \cdot \frac{1 - \alpha^{2l}}{1 - \alpha^2} + L\eta_t^2 p_{\max} \mathbf{G}^2
$$
$$
\overset{1)}{\leq} \frac{2D}{W} \cdot \frac{(1 - \alpha^l)}{1 - \alpha} + \frac{D}{W} \left[ (1 + \alpha^{l-1}) + \frac{(1 - \alpha^{l-1})(1 + \alpha)}{1 - \alpha} \right]
$$
$$
+ L\eta_t^2 p_{\max} \cdot \frac{\sigma^2}{W^2} \cdot \frac{1 - \alpha^{2l}}{1 - \alpha^2} + L\eta_t^2 p_{\max} \mathbf{G}^2
$$

$$(19)$$

The remaining follow the same procedure as Aydore et al. (2019), for brevity, we use the lemma on their paper. For d) we apply the (Aydore et al., 2019, Lemma 3.3) and derive $\mathbb{E} S_{t,l,\alpha}(w_0^t, \mathbf{w}^t) -$

$\mathbb{E}S_{t+1,l,\alpha}(w_0^{t+1}, \mathbf{w}^{t+1}) \leq \frac{2D}{W} \cdot \frac{(1-\alpha^l)}{1-\alpha}$. For e) we apply the (Aydore et al., 2019, Lemma 3.2) and derive $\mathbb{E}S_{t+1,l,\alpha}(w_0^{t+1}, \mathbf{w}^{t+1}) - \mathbb{E}S_{t,l,\alpha}(w_0^{t+1}, \mathbf{w}^{t+1}) \leq \frac{D}{W}\left[(1+\alpha^{l-1}) + \frac{(1-\alpha^{l-1})(1+\alpha)}{1-\alpha}\right]$. 1) plugs in d) and e).

Divide both side by $(\eta_t - L\eta_t^2)p_{\min}$.

$$
\begin{aligned}
&\mathbb{E}\left\|\nabla S_{t,l,\alpha}(w_0^t, \mathbf{w}^t)\right\|^2 \\
&\leq \frac{\frac{2D}{W}\cdot\frac{(1-\alpha^l)}{1-\alpha} + \frac{D}{W}\left[(1+\alpha^{l-1}) + \frac{(1-\alpha^{l-1})(1+\alpha)}{1-\alpha}\right] + L\eta_t^2 p_{\max}\cdot\frac{\sigma^2}{W^2}\cdot\frac{1-\alpha^{2l}}{1-\alpha^2} + L\eta_t^2 p_{\max}\mathbf{G}^2}{(\eta_t - L\eta_t^2)p_{\min}} \\
&\overset{1)}{\leq} \frac{4D}{\eta_t W p_{\min}}\cdot\frac{(1-\alpha^l)}{1-\alpha} + \frac{2D}{\eta_t W p_{\min}}\left[(1+\alpha^{l-1}) + \frac{(1-\alpha^{l-1})(1+\alpha)}{1-\alpha}\right] \\
&\quad + 2L\eta_t\frac{p_{\max}}{p_{\min}}\cdot\frac{\sigma^2}{W^2}\cdot\frac{1-\alpha^{2l}}{1-\alpha^2} + 2L\eta_t\frac{p_{\max}}{p_{\min}}\mathbf{G}^2 \\
&= \frac{2D}{\eta_t W p_{\min}}\left[\frac{2(1-\alpha^l)}{1-\alpha} + (1+\alpha^{l-1}) + \frac{(1-\alpha^{l-1})(1+\alpha)}{1-\alpha}\right] \\
&\quad + 2L\eta_t\frac{p_{\max}}{p_{\min}}\cdot\frac{\sigma^2}{W^2}\cdot\frac{1-\alpha^{2l}}{1-\alpha^2} + 2L\eta_t\frac{p_{\max}}{p_{\min}}\mathbf{G}^2
\end{aligned}
\tag{20}
$$

where 1) note that when $\eta_t \leq \frac{1}{2L}$, $\eta_t - L\eta_t^2 \leq \frac{1}{2}\eta_t$,

Follow the proof of (Aydore et al., 2019, Theorem 3.4), as $\alpha \to 1^-$

$$
\begin{aligned}
&\lim_{\alpha\to 1^-} \mathbb{E}\left\|\nabla S_{t,l,\alpha}(w_0^t, \mathbf{w}^t)\right\|^2 \\
&\leq \frac{8D}{\eta_t W p_{\min}} + 2L\eta_t\frac{p_{\max}}{p_{\min}}\cdot\frac{\sigma^2}{W^2}\cdot\frac{1-\alpha^{2l}}{1-\alpha^2} + 2L\eta_t\frac{p_{\max}}{p_{\min}}\mathbf{G}^2 \\
&\leq \frac{1}{W p_{\min}}\left(\frac{8D}{\eta_t} + 2L\eta_t p_{\max}\sigma^2\right) + 2L\eta_t\frac{p_{\max}}{p_{\min}}\mathbf{G}^2
\end{aligned}
\tag{21}
$$

Summing from $t = 0, 1, ...T$ concludes the proof.

$$
\begin{aligned}
DLR_l(T) &= \sum_{t=0}^{t}\lim_{\alpha\to 1^-}\mathbb{E}\left\|\nabla S_{t,l,\alpha}(w_0^t, \mathbf{w}^t)\right\|^2 \\
&\leq \frac{T}{W p_{\min}}\left(\frac{8D}{\eta_t} + 2L\eta_t p_{\max}\sigma^2\right) + 2TL\eta_t\frac{p_{\max}}{p_{\min}}\mathbf{G}^2
\end{aligned}
\tag{22}
$$

$\square$

*Proof of Corollary 1:*

Select $l = T^{\frac{1}{2}}$ and $\eta_t = T^{-\frac{1}{4}}$, note that $\lim_{\alpha\to 1^-} W = \lim_{\alpha\to 1^-}\frac{1-\alpha^l}{1-\alpha} = l$.

$$
\begin{aligned}
&\sum_{t=0}^{t}\lim_{\alpha\to 1^-}\mathbb{E}\left\|\nabla S_{t,l,\alpha}(w_0^t, \mathbf{w}^t)\right\|^2 \\
&\overset{1)}{\leq} \frac{T}{l p_{\min}}\left(\frac{8D}{\eta_t} + 2L\eta_t p_{\max}\sigma^2\right) + 2TL\eta_t\frac{p_{\max}}{p_{\min}}\mathbf{G}^2 \\
&\overset{2)}{\leq} \frac{T}{\sqrt{T}p_{\min}}\left(8DT^{\frac{1}{4}} + 2LT^{-\frac{1}{4}}p_{\max}\sigma^2\right) + 2T^{\frac{3}{4}}L\eta_t\frac{p_{\max}}{p_{\min}}\mathbf{G}^2 \\
&= \frac{8D}{p_{\min}}T^{\frac{3}{4}} + \frac{2Lp_{\max}\sigma^2}{p_{\min}}T^{\frac{1}{4}} + \frac{2L\eta_t p_{\max}\mathbf{G}^2}{p_{\min}}T^{\frac{3}{4}} \\
&= \mathcal{O}(T^{\frac{3}{4}})
\end{aligned}
\tag{23}
$$

$\blacksquare$

## B  REGRET ANALYSIS FOR OGD-EVENT IN CONVEX CASE

In the convex case, OGD is already an efficient algorithm to solve the online learning problem with partial client activation.

We start by defining Regret in the VFL framework below.

**Definition 3.** *Regret for online convex optimization in VFL framework*

$$R_T = \sum_{t=1}^{T} f(w_0^t, \mathbf{w}_0^t) - \sum_{t=1}^{T} f(w_0^*, \mathbf{w}^*) \tag{24}$$

*where* $[w_0*, \mathbf{w}^*] = \underset{[w_0^*, \mathbf{w}^*]}{argmin} \sum_{t=1}^{T} f(w_0^*, \mathbf{w}^*)$

Following the online convex optimization (Hazan et al., 2016), we design the event-driven online VFL with online gradient descent (OGD-Event in Section 5.1). The algorithm is provided in the algorithm 2 below.

---

**Algorithm 2** Event-driven online VFL with online gradient descent (OGD-Event)

---

**Input:**
**Output:** model parameter $w_m$ for all workers $m \in \{0, 1, ...M\}$.
  0: Initialize $w_m$ for all participants $m \in \{0, 1, ...M\}$
  1: **for** $t \in [T]$ **do**
  2:    **for** $m \in A_t$ **do**
  3:       client $m$ send $h_m(w_m; x_m)$ to the server.
  4:    **end for**
  5:    **for** $m \in \bar{A}_t$ **do**
  6:       Server queries the embeddings from passive client $m$.
  7:       Client $m$ send $h_m(w_m; x_m)$ to the server.
  8:    **end for**
  9:    The server updates its model $w_0 \leftarrow w_0 - \eta_0 \cdot \frac{\partial f(w_0, \mathbf{w})}{\partial w_0}$
 10:    **for** $m \in A_t$ **do**
 11:       Server send $v_m = \frac{\partial f(w_0, \mathbf{w})}{\partial h_m}$ to the client $m$.
 12:       Client $m$ update parameter $w_m \leftarrow w_m - \eta_m v_m \cdot \frac{\partial h_m}{\partial w_m}$
 13:    **end for**
 14: **end for**

---

We use a different set of assumptions on convexity, the diameter of the space, and the client's delay to make it fit the regret analysis framework for online convex optimization with Algorithm 2.

**Assumption 6. *Convexity:*** *for any* $[w_0, \mathbf{w}]$ *and* $[w_0', \mathbf{w}']$, $w_0, w_0' \in \mathbb{R}^{d_0}$, $w_m, w_m' \in R^{d_m}$, $f^t(w_0, \mathbf{w})$ *satisfy*

$$f^t(w_0', \mathbf{w}') \geq f^t(w_0, \mathbf{w}) + \left\langle \nabla f^t(w_0, \mathbf{w}), [w_0', \mathbf{w}'] - [w_0, \mathbf{w}] \right\rangle \tag{25}$$

**Assumption 7. *Bounded space diameter:*** *For any* $[w_0, \mathbf{w}]$ *and* $[w_0', \mathbf{w}']$, *satisfy*

$$\|[w_0, \mathbf{w}] - [w_0', \mathbf{w}']\| \leq D \tag{26}$$

**Assumption 8. Independent client activation:** *The activated client* $m_t$ *for the global iteration* $t$ *is independent of* $m_0, \cdots, m_{t-1}$ *and satisfies* $\mathbb{P}(m_t = m) := p_m$.

**Assumption 9. Uniformly bounded delay:** *For each client* $m$, *the delay at each global iteration* $t$ *is bounded by a constant* $\tau$. *i.e.* $\tau_m^t \leq \tau$.

**Theorem 2.** *Under Assumptions* $6 \sim 9$, *to solve the online VFL problem with partial client participation using Algorithm 2, the following inequality holds.*

$$R_T \leq \frac{3\mathbf{G}^2 D + 4D^3 + 2L^2\tau^2\mathbf{G}^2 D}{2\mathbf{G}\min\{p_m\}} \cdot \sqrt{T} \tag{27}$$

*where the learning rate is chosen as* $\eta_t = \frac{D}{\mathbf{G}\sqrt{t}}$.

*Remark* 3. The regret is sublinear $O(\sqrt{T})$.

*Proof of Theorem 2:*

First, we bound the update of the participants. For notation brevity, we use $\nabla_{w_0} f(w_0, \mathbf{w})$ and $\nabla_{w_m} f(w_0, \mathbf{w})$ are the partial derivative w.r.t. the corresponding parameter in the space of the parameter of the global model, where the position of all other parameters are filled with $0$.

$$\left\| [w_0^{t+1}, \mathbf{w}^{t+1}] - [w_0^*, \mathbf{w}^*] \right\|^2$$

$$\overset{1)}{=} \left\| [w_0^t, \mathbf{w}^t] - \eta_t \nabla_{w_0} f(w_0^t, \tilde{\mathbf{w}}^t) - \eta_t \sum_{m \in A_t} \nabla_{w_m} f(w_0^t, \tilde{\mathbf{w}}^t) - [w_0^*, \mathbf{w}^*] \right\|^2$$

$$\overset{2)}{=} \left\| [w_0^t, \mathbf{w}^t] - [w_0^*, \mathbf{w}^*] \right\|^2 + \eta_t^2 \left\| \nabla_{w_0} f(w_0, \tilde{\mathbf{w}}) + \sum_{m \in A_t} \nabla_{w_m} f(w_0^t, \tilde{\mathbf{w}}^t) \right\|^2$$

$$- 2\eta_t \left\langle \nabla_{w_0} f(w_0^t, \tilde{\mathbf{w}}^t) + \sum_{m \in A_t} \nabla_{w_m} f(w_0^t, \tilde{\mathbf{w}}^t), [w_0^t, \mathbf{w}^t] - [w_0^*, \mathbf{w}^*] \right\rangle$$

$$\overset{3)}{\leq} \left\| [w_0^t, \mathbf{w}^t] - [w_0^*, \mathbf{w}^*] \right\|^2 + \eta_t^2 \mathbf{G}^2$$

$$- 2\eta_t \left\langle \nabla_{w_0} f(w_0^t, \tilde{\mathbf{w}}^t) + \sum_{m \in A_t} \nabla_{w_m} f(w_0^t, \tilde{\mathbf{w}}^t), [w_0^t, \mathbf{w}^t] - [w_0^*, \mathbf{w}^*] \right\rangle$$

$$= \left\| [w_0^t, \mathbf{w}^t] - [w_0^*, \mathbf{w}^*] \right\|^2 + \eta_t^2 \mathbf{G}^2$$
$$- 2\eta_t \left\langle \nabla_{w_0} f(w_0^t, \tilde{\mathbf{w}}^t), [w_0^t, \mathbf{w}^t] - [w_0^*, \mathbf{w}^*] \right\rangle$$
$$- 2\eta_t \left\langle \sum_{m \in A_t} \nabla_{w_m} f(w_0^t, \tilde{\mathbf{w}}^t), [w_0^t, \mathbf{w}^t] - [w_0^*, \mathbf{w}^*] \right\rangle$$

$$= \left\| [w_0^t, \mathbf{w}^t] - [w_0^*, \mathbf{w}^*] \right\|^2 + \eta_t^2 \mathbf{G}^2$$
$$- 2\eta_t \left\langle \nabla_{w_0} f(w_0^t, \tilde{\mathbf{w}}^t) - \nabla_{w_0} f(w_0^t, \mathbf{w}^t) + \nabla_{w_0} f(w_0^t, \mathbf{w}^t), [w_0^t, \mathbf{w}^t] - [w_0^*, \mathbf{w}^*] \right\rangle$$
$$- 2\eta_t \left\langle \sum_{m \in A_t} \nabla_{w_m} f(w_0^t, \tilde{\mathbf{w}}^t) - \sum_{m \in A_t} \nabla_{w_m} f(w_0^t, \mathbf{w}^t) + \sum_{m \in A_t} \nabla_{w_m} f(w_0^t, \mathbf{w}^t), [w_0^t, \mathbf{w}^t] - [w_0^*, \mathbf{w}^*] \right\rangle$$

$$= \left\| [w_0^t, \mathbf{w}^t] - [w_0^*, \mathbf{w}^*] \right\|^2 + \eta_t^2 \mathbf{G}^2$$
$$+ 2 \left\langle \nabla_{w_0} f(w_0^t, \mathbf{w}^t) - \nabla_{w_0} f(w_0^t, \tilde{\mathbf{w}}^t), \eta_t \left( [w_0^t, \mathbf{w}^t] - [w_0^*, \mathbf{w}^*] \right) \right\rangle - 2\eta_t \left\langle \nabla_{w_0} f(w_0^t, \mathbf{w}^t), [w_0^t, \mathbf{w}^t] - [w_0^*, \mathbf{w}^*] \right\rangle$$
$$+ 2 \left\langle \sum_{m \in A_t} \nabla_{w_m} f(w_0^t, \mathbf{w}^t) - \sum_{m \in A_t} \nabla_{w_m} f(w_0^t, \tilde{\mathbf{w}}^t), \eta_t \left( [w_0^t, \mathbf{w}^t] - [w_0^*, \mathbf{w}^*] \right) \right\rangle$$
$$- 2\eta_t \left\langle \sum_{m \in A_t} \nabla_{w_m} f(w_0^t, \mathbf{w}^t), [w_0^t, \mathbf{w}^t] - [w_0^*, \mathbf{w}^*] \right\rangle$$

$$\overset{4)}{\leq} \left\| [w_0^t, \mathbf{w}^t] - [w_0^*, \mathbf{w}^*] \right\|^2 + \eta_t^2 \mathbf{G}^2$$
$$+ \left\| \nabla_{w_0} f(w_0^t, \mathbf{w}^t) - \nabla_{w_0} f(w_0^t, \tilde{\mathbf{w}}^t) \right\|^2 + \eta_t^2 \left\| ([w_0^t, \mathbf{w}^t] - [w_0^*, \mathbf{w}^*]) \right\|^2$$
$$- 2\eta_t \left\langle \nabla_{w_0} f(w_0^t, \mathbf{w}^t), [w_0^t, \mathbf{w}^t] - [w_0^*, \mathbf{w}^*] \right\rangle$$
$$+ \left\| \sum_{m \in A_t} \nabla_{w_m} f(w_0^t, \mathbf{w}^t) - \sum_{m \in A_t} \nabla_{w_m} f(w_0^t, \tilde{\mathbf{w}}^t) \right\|^2 + \eta_t^2 \left\| ([w_0^t, \mathbf{w}^t] - [w_0^*, \mathbf{w}^*]) \right\|^2$$
$$- 2\eta_t \left\langle \sum_{m \in A_t} \nabla_{w_m} f(w_0^t, \mathbf{w}^t), [w_0^t, \mathbf{w}^t] - [w_0^*, \mathbf{w}^*] \right\rangle$$

$$
\stackrel{5)}{=} \left\| [w_0^t, \mathbf{w}^t] - [w_0^*, \mathbf{w}^*] \right\|^2 + \eta_t^2 \mathbf{G}^2
$$

$$
+ \left\| \nabla_{w_0} f(w_0^t, \mathbf{w}^t) + \sum_{m \in A_t} \nabla_{w_m} f(w_0^t, \mathbf{w}^t) - \nabla_{w_0} f(w_0^t, \tilde{\mathbf{w}}^t) - \sum_{m \in A_t} \nabla_{w_m} f(w_0^t, \tilde{\mathbf{w}}^t) \right\|^2
$$

$$
+ \eta_t^2 \left\| ([w_0^t, \mathbf{w}^t] - [w_0^*, \mathbf{w}^*]) \right\|^2 - 2\eta_t \left\langle \nabla_{w_0} f(w_0^t, \mathbf{w}^t), [w_0^t, \mathbf{w}^t] - [w_0^*, \mathbf{w}^*] \right\rangle
$$

$$
+ \eta_t^2 \left\| ([w_0^t, \mathbf{w}^t] - [w_0^*, \mathbf{w}^*]) \right\|^2 - 2\eta_t \left\langle \sum_{m \in A_t} \nabla_{w_m} f(w_0^t, \mathbf{w}^t), [w_0^t, \mathbf{w}^t] - [w_0^*, \mathbf{w}^*] \right\rangle
$$

$$
\stackrel{6)}{\leq} \left\| [w_0^t, \mathbf{w}^t] - [w_0^*, \mathbf{w}^*] \right\|^2 + \eta_t^2 \mathbf{G}^2
$$

$$
+ L^2 \left\| \mathbf{w}^t - \tilde{\mathbf{w}}^t \right\|^2
$$

$$
+ 2\eta_t^2 \left\| ([w_0^t, \mathbf{w}^t] - [w_0^*, \mathbf{w}^*]) \right\|^2
$$

$$
- 2\eta_t \left\langle \nabla_{w_0} f(w_0^t, \mathbf{w}^t), [w_0^t, \mathbf{w}^t] - [w_0^*, \mathbf{w}^*] \right\rangle - 2\eta_t \left\langle \sum_{m \in A_t} \nabla_{w_m} f(w_0^t, \mathbf{w}^t), [w_0^t, \mathbf{w}^t] - [w_0^*, \mathbf{w}^*] \right\rangle
$$

$$
\stackrel{7)}{\leq} \left\| [w_0^t, \mathbf{w}^t] - [w_0^*, \mathbf{w}^*] \right\|^2 + \eta_t^2 \mathbf{G}^2 + L^2 \tau^2 \mathbf{G}^2 \max_{i \in [\tau]} \left\{ \eta_{t-i}^2 \right\}
$$

$$
+ 2\eta_t^2 \left\| ([w_0^t, \mathbf{w}^t] - [w_0^*, \mathbf{w}^*]) \right\|^2
$$

$$
- 2\eta_t \left\langle \nabla_{w_0} f(w_0^t, \mathbf{w}^t), [w_0^t, \mathbf{w}^t] - [w_0^*, \mathbf{w}^*] \right\rangle - 2\eta_t \left\langle \sum_{m \in A_t} \nabla_{w_m} f(w_0^t, \mathbf{w}^t), [w_0^t, \mathbf{w}^t] - [w_0^*, \mathbf{w}^*] \right\rangle
$$

$$
\stackrel{8)}{\leq} \left\| [w_0^t, \mathbf{w}^t] - [w_0^*, \mathbf{w}^*] \right\|^2 + \eta_t^2 \mathbf{G}^2 + L^2 \tau^2 \mathbf{G}^2 \max_{i \in [\tau]} \left\{ \eta_{t-i}^2 \right\} + 2\eta_t^2 D^2
$$

$$
- 2\eta_t \left\langle \nabla_{w_0} f(w_0^t, \mathbf{w}^t), [w_0^t, \mathbf{w}^t] - [w_0^*, \mathbf{w}^*] \right\rangle - 2\eta_t \left\langle \sum_{m \in A_t} \nabla_{w_m} f(w_0^t, \mathbf{w}^t), [w_0^t, \mathbf{w}^t] - [w_0^*, \mathbf{w}^*] \right\rangle
$$

$$(28)$$

where 1) is the partial activation of clients and server optimization step at time step $t$, 2) note that $\{\nabla_{w_m} f(w_0^t, \tilde{\mathbf{w}}^t)\}_{m \in \{0\} \cup A_t}$ are in the non-intersect dimensions, 3) applies assumption 4 (bounded gradient), 4) applies $\langle a, b \rangle \leq \frac{1}{2} \|a\|^2 + \frac{1}{2} \|b\|^2$, 5) $\{\nabla_{w_m} f(w_0^t, \tilde{\mathbf{w}}^t)\}_{m \in \{0\} \cup A_t}$ are in the non-intersect dimensions, 6) applies assumption assumption 1 (Lipschitz Gradient). 7) applies Eq. 29 below, 8) applies assumption 7 (bounded parameter diameter).

$$
\left\| \mathbf{w}^t - \tilde{\mathbf{w}}^t \right\|^2
$$

$$
\stackrel{1)}{\leq} \left\| \sum_{i=1}^{\tau} \left( \mathbf{w}^{t+1-i} - \mathbf{w}^{t-i} \right) \right\|^2
$$

$$
\stackrel{2)}{\leq} \tau \sum_{i=1}^{\tau} \left\| \mathbf{w}^{t+1-i} - \mathbf{w}^{t-i} \right\|^2
$$

$$
= \tau \sum_{i=1}^{\tau} \left\| -\eta_{t-i} \sum_{m \in A_{t-i}} \nabla_{w_m} f(w_0^{t-i}, \mathbf{w}^{t-i}) \right\|^2
$$

$$
= \tau \sum_{i=1}^{\tau} \eta_{t-i}^2 \left\| \sum_{m \in A_{t-i}} \nabla_{w_m} f(w_0^{t-i}, \mathbf{w}^{t-i}) \right\|^2
$$

$$
\stackrel{3)}{\leq} \tau \mathbf{G}^2 \sum_{i=1}^{\tau} \eta_{t-i}^2
$$

$$\leq \tau^2 \mathbf{G}^2 \max_{i \in [\tau]} \left\{ \eta_{t-i}^2 \right\} \tag{29}$$

where 1) applies assumption 9 (uniformly bounded delay), 2) by Cauchy-Schwarz inequality, $\left( \sum_{i=0}^{n-1} x_i \right)^2 = \left( \sum_{i=0}^{n-1} 1 \cdot x_i \right)^2 \leq n \sum_{i=0}^{n-1} x_i^2$, 3) applies assumption 4 (bounded gradient), noting that the gradients are in non-intersect dimensions.

Rearrange Eq. 28:

$$2\eta_t \left\langle \nabla_{w_0} f(w_0^t, \mathbf{w}^t), [w_0^t, \mathbf{w}^t] - [w_0^*, \mathbf{w}^*] \right\rangle + 2\eta_t \left\langle \sum_{m \in A_t} \nabla_{w_m} f(w_0^t, \mathbf{w}^t), [w_0^t, \mathbf{w}^t] - [w_0^*, \mathbf{w}^*] \right\rangle$$

$$\leq \left\| [w_0^t, \mathbf{w}^t] - [w_0^*, \mathbf{w}^*] \right\|^2 - \left\| [w_0^{t+1}, \mathbf{w}^{t+1}] - [w_0^*, \mathbf{w}^*] \right\|^2 + \eta_t^2 \mathbf{G}^2 + L^2 \tau^2 \mathbf{G}^2 \max_{i \in [\tau]} \left\{ \eta_{t-i}^2 \right\} + 2\eta_t^2 D^2 \tag{30}$$

Taking expectation w.r.t. the activation client sets $A_t$ from both sides.

$$2\eta_t \left\langle \nabla_{w_0} f(w_0^t, \mathbf{w}^t), [w_0^t, \mathbf{w}^t] - [w_0^*, \mathbf{w}^*] \right\rangle + 2\eta_t \left\langle \sum_{m=1}^{M} p_m \nabla_{w_m} f(w_0^t, \mathbf{w}^t), [w_0^t, \mathbf{w}^t] - [w_0^*, \mathbf{w}^*] \right\rangle$$

$$\leq \mathbb{E} \left\| [w_0^t, \mathbf{w}^t] - [w_0^*, \mathbf{w}^*] \right\|^2 - \mathbb{E} \left\| [w_0^{t+1}, \mathbf{w}^{t+1}] - [w_0^*, \mathbf{w}^*] \right\|^2 + \eta_t^2 \mathbf{G}^2 + L^2 \tau^2 \mathbf{G}^2 \max_{i \in [\tau]} \left\{ \eta_{t-i}^2 \right\} + 2\eta_t^2 D^2 \tag{31}$$

Taking the minimum of $p_m$:

$$2\eta_t \min \{p_m\} \left\langle \nabla_{w_0} f(w_0^t, \mathbf{w}^t), [w_0^t, \mathbf{w}^t] - [w_0^*, \mathbf{w}^*] \right\rangle + 2\eta_t \min \{p_m\} \left\langle \sum_{m=1}^{M} \nabla_{w_m} f(w_0^t, \mathbf{w}^t), [w_0^t, \mathbf{w}^t] - [w_0^*, \mathbf{w}^*] \right\rangle$$

$$\leq \mathbb{E} \left\| [w_0^t, \mathbf{w}^t] - [w_0^*, \mathbf{w}^*] \right\|^2 - \mathbb{E} \left\| [w_0^{t+1}, \mathbf{w}^{t+1}] - [w_0^*, \mathbf{w}^*] \right\|^2 + \eta_t^2 \mathbf{G}^2 + L^2 \tau^2 \mathbf{G}^2 \max_{i \in [\tau]} \left\{ \eta_{t-i}^2 \right\} + 2\eta_t^2 D^2 \tag{32}$$

Combining the gradient from different dimensions, and rearranging the equation:

$$\left\langle \nabla_{[w_0, \mathbf{w}]} f(w_0^t, \mathbf{w}^t), [w_0^t, \mathbf{w}^t] - [w_0^*, \mathbf{w}^*] \right\rangle$$

$$\leq \frac{1}{2\eta_t \min \{p_m\}} \left( \mathbb{E} \left\| [w_0^t, \mathbf{w}^t] - [w_0^*, \mathbf{w}^*] \right\|^2 - \mathbb{E} \left\| [w_0^{t+1}, \mathbf{w}^{t+1}] - [w_0^*, \mathbf{w}^*] \right\|^2 \right)$$

$$+ \frac{1}{2\eta_t \min \{p_m\}} \left( \eta_t^2 \mathbf{G}^2 + L^2 \tau^2 \mathbf{G}^2 \max_{i \in [\tau]} \left\{ \eta_{t-i}^2 \right\} + 2\eta_t^2 D^2 \right)$$

$$\overset{1)}{\leq} \frac{1}{2 \min \{p_m\}} \left( \frac{Q}{\eta_t} + \eta_t \mathbf{G}^2 + \frac{L^2 \tau^2 \mathbf{G}^2 \max_{i \in [\tau]} \left\{ \eta_{t-i}^2 \right\}}{\eta_t} + 2\eta_t D^2 \right) \tag{33}$$

where 1) denote $Q = \mathbb{E} \left\| [w_0^t, \mathbf{w}^t] - [w_0^*, \mathbf{w}^*] \right\|^2 - \mathbb{E} \left\| [w_0^{t+1}, \mathbf{w}^{t+1}] - [w_0^*, \mathbf{w}^*] \right\|^2$ for brevity.

By convexity (assumption 6):

$$f^t(w_0^t, \mathbf{w}^t) - f^t(w_0^*, \mathbf{w}^*) \leq \left\langle \nabla_{[w_0, \mathbf{w}]} f^t, [w_0^t, \mathbf{w}^t] - [w_0^*, \mathbf{w}^*] \right\rangle \tag{34}$$

Summing over $t = 1 \ldots T$, and if we set a diminish learning rate $\eta_t = \frac{D}{\mathbf{G}\sqrt{t}}$ with $(\frac{1}{\eta_0} \triangleq 0)$:

$$\sum_{t=1}^{T} \left( f^t(w_0^t, \mathbf{w}^t) - f^t(w_0^*, \mathbf{w}^*) \right)$$

$$\leq \sum_{t=1}^{T} \left( \left\langle \nabla_{[w_0, \mathbf{w}]} f^t, [w_0^t, \mathbf{w}^t] - [w_0^*, \mathbf{w}^*] \right\rangle \right)$$

$$\overset{1)}{\leq} \frac{1}{2\min\{p_m\}} \sum_{t=1}^{T} \left( \frac{Q}{\eta_t} + \eta_t \mathbf{G}^2 + \frac{L^2\tau^2\mathbf{G}^2 \max_{i\in[\tau]}\{\eta_{t-i}^2\}}{\eta_t} + 2\eta_t D^2 \right)$$

$$\overset{2)}{=} \frac{1}{2\min\{p_m\}} \left\{ \sum_{t=1}^{T} \frac{\mathbb{E}\left\| [w_0^t, \mathbf{w}^t] - [w_0^*, \mathbf{w}^*] \right\|^2 - \mathbb{E}\left\| [w_0^{t+1}, \mathbf{w}^{t+1}] - [w_0^*, \mathbf{w}^*] \right\|^2}{\eta_t} + \sum_{t=1}^{T} \left[ \eta_t(\mathbf{G}^2 + 2D^2 + L^2\tau^2\mathbf{G}^2) \right] \right\}$$

$$\overset{3)}{\leq} \frac{1}{2\min\{p_m\}} \left\{ \sum_{t=1}^{T} \mathbb{E}\left\| [w_0^t, \mathbf{w}^t] - [w_0^*, \mathbf{w}^*] \right\|^2 \left( \frac{1}{\eta_t} - \frac{1}{\eta_{t+1}} \right) + \sum_{t=1}^{T} \left[ \eta_t(\mathbf{G}^2 + 2D^2 + L^2\tau^2\mathbf{G}^2) \right] \right\}$$

$$\overset{4)}{\leq} \frac{1}{2\min\{p_m\}} \left\{ \sum_{t=1}^{T} D^2 \left( \frac{1}{\eta_t} - \frac{1}{\eta_{t+1}} \right) + \sum_{t=1}^{T} \left[ \eta_t(\mathbf{G}^2 + 2D^2 + L^2\tau^2\mathbf{G}^2) \right] \right\}$$

$$= \frac{1}{2\min\{p_m\}} \left[ D^2 \frac{1}{\eta_T} + (\mathbf{G}^2 + 2D^2 + L^2\tau^2\mathbf{G}^2) \sum_{t=1}^{T} \eta_t \right]$$

$$\overset{5)}{\leq} \frac{1}{2\min\{p_m\}} \left[ \mathbf{G}D\sqrt{T} + (\mathbf{G}D + \frac{2D^3}{\mathbf{G}} + L^2\tau^2\mathbf{G}D) \cdot 2\sqrt{T} \right]$$

$$\leq \frac{3\mathbf{G}^2 D + 4D^3 + 2L^2\tau^2\mathbf{G}^2 D}{2\mathbf{G}\min\{p_m\}} \sqrt{T} \tag{35}$$

where 1) applies Eq. 33,
2) $\eta_t$ is diminish. Expand $Q$,
3) $\frac{1}{\eta_0} \triangleq 0$, and $\left\| [w_0^{T+1}, \mathbf{w}^{T+1}] - [w_0^*, \mathbf{w}^*] \right\|^2 \geq 0$,
4) applies assumption 7 (bounded parameter diameter).
5) $\sum_{t=1}^{T} \frac{1}{\sqrt{t}} \leq 2\sqrt{T}$.

The proof of Theorem 2 is complete. ∎

## C  EXTRA DETAILS

### C.1  ALGORITHM 1 IN A SYNCHRONOUS MANNER

We also provide a synchronous version of the algorithm 1 below.

---

**Algorithm 3** Event-driven online VFL on Dynamic Local Regret

---

**Input:** hyperparameter $l, \alpha, \eta$
**Output:** server model $w_0$, client models $w_m \in [M]$
  0: initialize model $w_m$ for all participants $m \in \{0, 1, ...M\}$
  1: **for** $t \in [T]$ **do**
  2:   **for** $m \in A_t$ **do**
  3:     activated client $m$ send $h_m(w_m^t; x_m^t)$ to the server.
  4:   **end for**
  5:   **for** $m \in \bar{A}_t$ **do**
  6:     server queries the passive client $m$.
  7:     passive client $m$ send $h_m(w_m^t; x_m^t)$ to the server.
  8:   **end for**
  9:   server updates its model $w_0^{t+1} \leftarrow w_0^t - \eta_0 \cdot \nabla_{w_0} S_{t,l,\alpha}(w_0^t, \mathbf{w}^t)$
 10:   **for** $m \in A_t$ **do**
 11:     server send $v_m = \frac{\partial S_{t,l,\alpha}(w_0^t, \mathbf{w}^t)}{\partial h_m(w_m^t; x_m^t)}$ to the client $m$.
 12:     client $m$ update parameter $w_m^{t+1} \leftarrow w_m^t - \eta_m \cdot v_m \cdot \frac{\partial h_m}{\partial w_m}$
 13:   **end for**
 14: **end for**

---

## C.2 ALGORITHM: OGD-FULL

Below we introduce the algorithm for the OGD-Full baseline in Algorithm 4 adapted from Wang & Xu (2023), and transformed into a partial gradient-based approach similar to the split neural network method proposed by Vepakomma et al. (2018). It is important to highlight that the utilization of multiple local updates in Wang & Xu (2023) is not applicable within the context of our study on the general VFL framework because the multiple local update approach requires that clients possess labeled data, which does not align with the setting in our paper.

---

**Algorithm 4** Online VFL with Online Gradient Descent (OGD-Full)

---

**Input:** Learning rate $\eta$
**Output:** Model parameter $w_m$ for all workers $m \in \{0, 1, ...M\}$.
  0: Initialize $w_m$ for all participants $m \in \{0, 1, ...M\}$
  1: **for** $t \in [T]$ **do**
  2:     **for** $m \in [M]$ **do**
  3:         Client $m$ send $h_m(w_m; x_m)$ to the server.
  4:     **end for**
  5:     The server updates its model $w_0 \leftarrow w_0 - \eta_0 \cdot \frac{\partial f(w_0, \mathbf{w})}{\partial w_0}$
  6:     **for** $m \in [M]$ **do**
  7:         Server send $v_m = \frac{\partial f(w_0, \mathbf{w})}{\partial h_m}$ to the client $m$.
  8:         Client $m$ update parameter $w_m \leftarrow w_m - \eta_m v_m \cdot \frac{\partial h_m}{\partial w_m}$
  9:     **end for**
 10: **end for**

---

## C.3 ALGORITHM: EVENT-DRIVEN ONLINE VFL USING STATIC LOCAL REGRET

We also provide a Static Time-Smoothed Stochastic Gradient Descent algorithm (Aydore et al., 2019; Hazan et al., 2017) which is adapted to Event-Driven Online VFL. Note that the static slide window is $F_{t,l}(w_0^t, \mathbf{w}^t) = \frac{1}{l} \sum_{i=0}^{l} f^{t-i}(w_0^t, \mathbf{w}^t)$, where the time step for the parameter is $t$.

---

**Algorithm 5** Event-driven online VFL on Static Local Regret

---

**Input:** Hyperparameter $l$, $\alpha$, $\eta$, fixed model $w^*$
**Output:** Server model $w_0$, client models $w_m \in [M]$
  0: Initialize model $w_m$ for all participants $m \in \{0, 1, \ldots, M\}$
  1: **for** $t \in [T]$ **do**
  2:     **for** $m \in A_t$ **do**
  3:         Activated client $m$ sends a window of embeddings $\{h_m(w_m^t; x_m^{t-i})\}_{i=0}^{l-1}$ to the server.
  4:     **end for**
  5:     **for** $m \in \bar{A}_t$ **do**
  6:         Server queries the passive client $m$, for the window of embeddings of length $l$
  7:         Passive clients $m$ sends the window of embeddings $\{h_m(w_m^t; x_m^{t-i})\}_{i=0}^{l-1}$ to the server.
  8:     **end for**
  9:     Server updates its model $w_0^{t+1} \leftarrow w_0^t - \eta_0 \cdot \nabla_{w_0} F_{t,l}(w_0^t, \mathbf{w}^t)$
 10:     # $\nabla_{w_0} F_{t,l}(w_0^t, \mathbf{w}^t) = \frac{1}{l} \sum_{i=0}^{l} \nabla_{w_0} f^{t-i}(w_0^t, \mathbf{w}^t)$
 11:     **for** $m \in A_t$ **do**
 12:         Server sends a window of gradient $\{\nabla_{h_m} f^{t-i}(w_0^t, \mathbf{w}^t)\}_{i=0}^{l-1}$ to client $m$.
 13:         Client $m$ updates parameter $w_m^{t+1} \leftarrow w_m^t - \eta_m \cdot \nabla_{w_m} F_{t,l}(w_0^t, \mathbf{w}^t)$
 14:         #    $\nabla_{w_m} F_{t,l}(w_0^t, \mathbf{w}^t)$    =    $\frac{1}{l} \sum_{i=0}^{l} \nabla_{w_m} f^{t-i}(w_0^t, \mathbf{w}^t)$    = $\frac{1}{l} \sum_{i=0}^{l} \nabla_{h_m} f^{t-i}(w_0^t, \mathbf{w}^t) \cdot \nabla_{w_m} h_m(w_m^t; x_m^{t-i})$
 15:     **end for**
 16: **end for**

---

# D    SUPPLEMENT EXPERIMENTS

## D.1    ABLATION STUDY OF DLR PARAMETERS

In order to achieve a thorough comprehension of applying DLR in the VFL framework, we conduct an ablation study on several key parameters, including the sliding window length $l$, and attenuation coefficient $\alpha$.

**Sliding window length** $l$    The sliding window length $l$ determines the length of the exponential average window of the DLR. A larger window implies that a greater number of past gradients influence the update at the current time. In the DLR-Full framework, employed under the concept drift data stream and with an attenuation coefficient of $\alpha = 0.95$, we vary the window length $l = 10, 50, 100, 150$, and conduct the training. It's worth noting that $0.95^{10} \approx 0.60$, $0.95^{50} \approx 0.08$, and $0.95^{100} \approx 0.006$ represent three different shapes of the exponential sliding average window.

Figure 5 presents the run-time error rate of different window lengths in the DLR-Full framework. As observed in the figure, the DLR is stable across different window lengths, which is consistent with the result reported by Aydore et al. (2019).

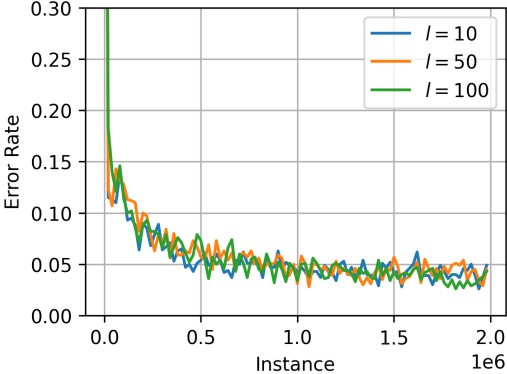

Figure 5: Ablation study on window length $l$

**Attenuation coefficient** $\alpha$    The attenuation coefficient of dynamic local regret influences the weighting of past gradients: a larger $\alpha$ indicates that the influence of past gradients decreases more slowly, while a smaller $\alpha$ leads to a quicker attenuation of the effect of old parameters. For our experiment, we utilize a sliding window length of 100 and vary $\alpha$ across 0.99, 0.95, and 0.9 in the DLR-Full framework. Notably, $0.99^{100} \approx 0.37$, $0.95^{90} \approx 0.01$, and $0.9^{44} \approx 0.01$ represent three distinct shapes of the exponential averaging window. Figure 6 illustrates the run-time error rate for different values of $\alpha$. We observed that better results were obtained for $\alpha = 0.95$ and 0.9. Therefore, we recommend using exponential average windows where the tail approximates 0.

## D.2    EXPERIMENT ON OTHER ONLINE LEARNING DATASET

**SUSY dataset**    SUSY (Whiteson, 2014b) is a physics dataset from the UCI repository. It is a classification problem to distinguish between a signal process that produces supersymmetric particles and a background process that does not. Eighteen features are used for each sample to determine whether it belongs to the signal or background class. The first 8 features are kinematic properties measured by the particle detectors in the accelerator. The last ten features are functions of the first 8 features; these are high-level features derived by physicists to help discriminate between the two classes. The preprocess method follows the same process in Sahoo et al. (2018). In the experiment, two clients are employed, and each holds half of the feature set. Each client models are composed of four fully-connected layers with the following neuron configurations: 32, 64, 98, and an output layer containing 128 neurons. After each hidden layer, a Rectified Linear Unit (ReLU) activation function is applied to introduce non-linearity. On the server side, a five-layer MLP is employed. This model integrates the concatenated outputs from the client models and processes them through

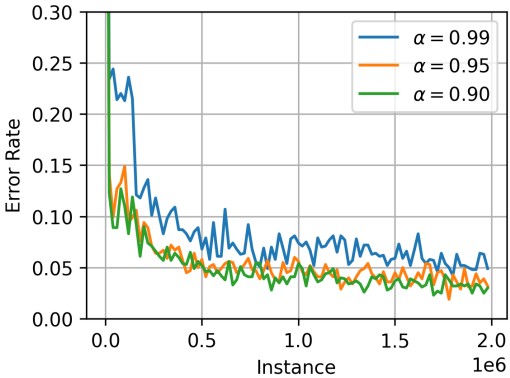

Figure 6: Attenuation coefficient $\alpha$

successive fully-connected layers with neuron counts of 256, 128, 64, and 32, ending with an output layer with 2 neurons. We tested various learning rates within the range of [0.01, 0.003, 0.001] to identify the optimal setting for our models under different experimental conditions. The exponential weighted sliding window's length of the DLR is tuned from $\{10, 50\}$, and the attenuation coefficient $\alpha$ from $\{0.95, 0.99\}$. The activation probability $p$ for the "Random" activation is selected from $\{0.25, 0.5, 0.75\}$. The activation threshold $\Gamma$ is tuned from $\{-1, -0.5, 0, 0.5\}$. The results for both the stationary and non-stationary data streams are depicted in Figures 7 and 8, respectively. These figures demonstrate that the DLR model achieves stable convergence in both Full and Partial activation modes, consistently outperforming OGD across varying data conditions. Additionally, DLR is learning faster than SLR in both scenarios.

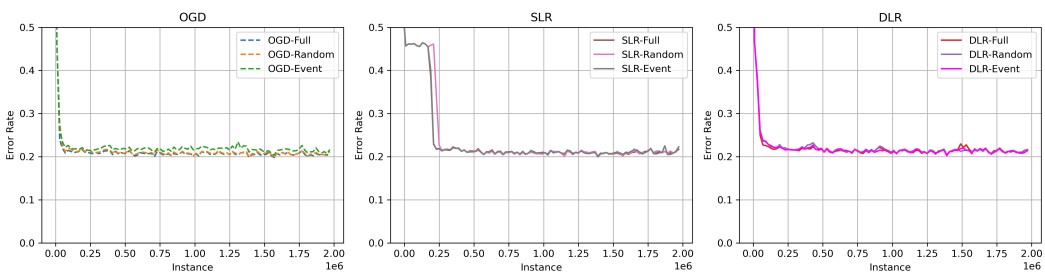

Figure 7: SUSY dataset, stationary case

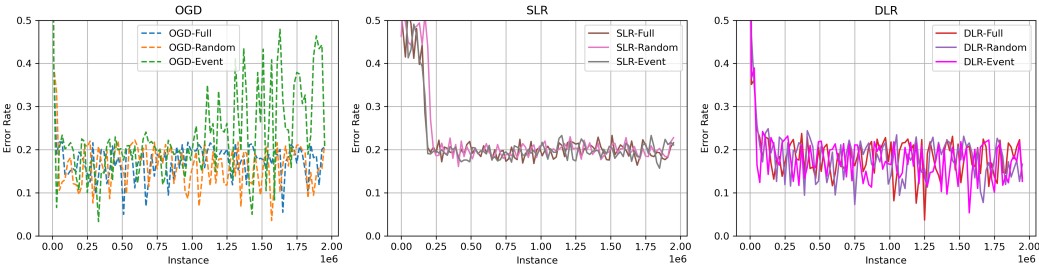

Figure 8: SUSY dataset, non-stationary case

**HIGGS dataset**    HIGGS (Whiteson, 2014a) is also a physics dataset from UCI repository. It is a classification challenge aimed at distinguishing between a signal process that produces Higgs bosons and a background process that does not. Each sample comprises 28 features: 21 low-level

features representing kinematic properties measured by particle detectors in the accelerator and 7 high-level features derived from the first 21. The model architecture employed in this experiment is identical to that used in the SUSY study, with the sole difference being the number of input features. The hyperparameter tuning procedure also remains consistent with the previous experiment. The results for both stationary and non-stationary conditions are depicted in Figure 9 and Figure 10, respectively. These results illustrate that the DLR model achieves comparable performance under stationary conditions and exhibits enhanced stability under non-stationary conditions. Furthermore, DLR demonstrates faster learning compared to SLR, with less initial stagnation at the early phases of training.

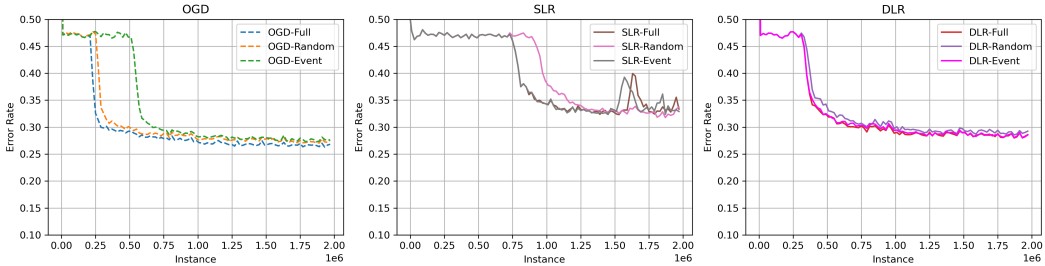

Figure 9: HIGGS dataset, stationary case

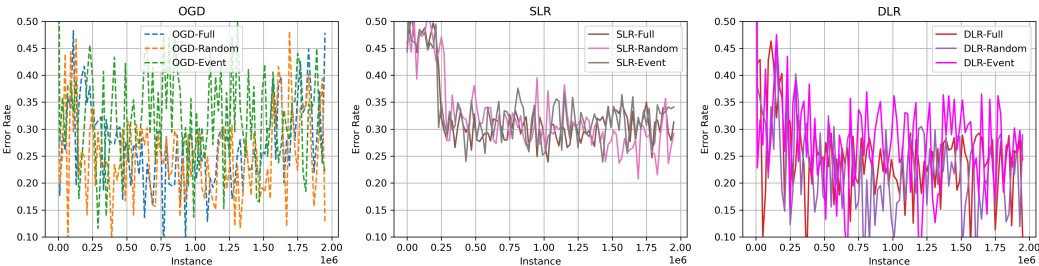

Figure 10: HIGGS dataset, non-stationary case

## D.3   ADDITIONAL EXPERIMENT ON EVENT-DRIVEN ONLINE VFL

**Scalability of the framework (number of total clients):**   To evaluate scalability, we increase the total number of clients in the experiment described in Section 5.2 on the stationary iMNIST dataset to 8 and 16 clients. Most settings remain consistent with those described in Section 5.1, with the number of clients increased to 8 and 16. Features are evenly distributed among the clients, and the input size of each client's model is adjusted accordingly. While the output size of the clients remains unchanged, the input size of the server model is scaled up to accommodate the increased number of clients. The results for 8 clients are presented in Figure 11, while the results for 16 clients are presented in Figure 12. Those figures demonstrate that when scale up the number of clients, the conclusion remain consistent: OGD is less stable under partial client activation, while DLR converges more rapidly than SLR. This consistency supports the robustness and scalability of our proposed framework.

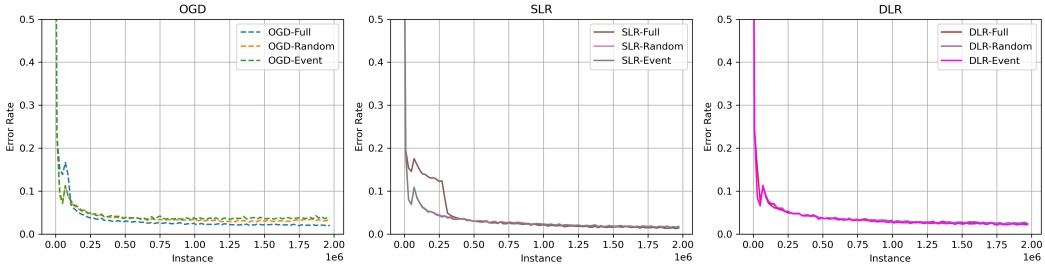

Figure 11: Run-time error rates on the iMNIST with 16 clients under stationary data stream

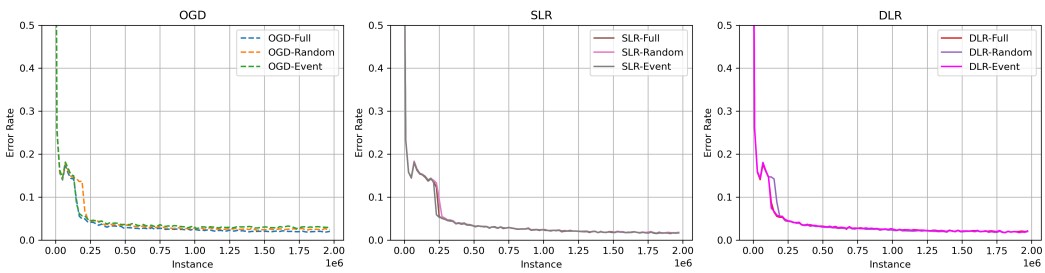

Figure 12: Run-time error rates on the iMNIST with 16 clients under stationary data stream

**Number of activated clients:** To further examine the impact of partial client activation, we conducted experiments in which a fixed number of clients were randomly selected to participate in each round. The experimental setup remains consistent with the configuration described in Section 5.1. The experiment involves 4 clients, with the activated clients randomly selected in each round. The number of activated clients varies from 1 to 4. The results are presented in Figure 13. The results suggest that increasing the number of activated clients enhances the stability of learning and reduces the variability in error rate over time. However, even with a single activated client, the model eventually achieves comparable performance, demonstrating robustness under limited client activation.

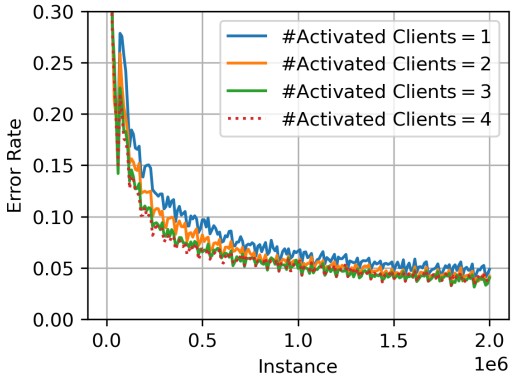

Figure 13: Number of activated client.

**Client activation statistic:** To further analyze and validate the partial client activation mechanism, we conducted additional experiments to monitor and record the activation frequency of each client throughout the training process. Table 5 presents the activation frequencies for each client under various event-driven framework settings described in Section 5.4, considering different activation probabilities $p$ and activation thresholds $\Gamma$. Note that the activation frequency for each client is defined as the ratio of the number of iterations in which the client was activated to the total number

of iterations. Additionally, we present results for extreme $\Gamma$ values, including $\Gamma = +\infty$ (where no clients are activated) and $\Gamma = -\infty$ (where all clients are always activated). This table also provides insight into the selection of $\Gamma$ values in Section 5.4, ensuring that the chosen $\Gamma$ values cover a wide range of activation frequencies for the clients.

Table 5: Frequency of activation for each client under different settings

| Setting | Client 1 | Client 2 | Client 3 | Client 4 |
|---|---|---|---|---|
| **Random** | | | | |
| $p = 0.25$ | 0.249747 | 0.250473 | 0.250277 | 0.250142 |
| $p = 0.5$ | 0.499507 | 0.499811 | 0.500266 | 0.500179 |
| $p = 0.75$ | 0.749623 | 0.750214 | 0.750155 | 0.750257 |
| $p = 1.0$ | 1.0 | 1.0 | 1.0 | 1.0 |
| **Event** | | | | |
| $\Gamma = +\infty$ $(+100)$ | 0 | 0 | 0 | 0 |
| $\Gamma = 0.6$ | 0.000006 | 0.066729 | 0.101827 | 0.000204 |
| $\Gamma = 0.2$ | 0.006346 | 0.461186 | 0.456317 | 0.038649 |
| $\Gamma = -0.2$ | 0.355166 | 0.982896 | 0.985216 | 0.596582 |
| $\Gamma = -\infty$ $(-100)$ | 1.0 | 1.0 | 1.0 | 1.0 |

# E    RELATED WORKS

## E.1    VERTICAL FEDERATED LEARNING

Federated learning is a decentralized machine learning approach where models are trained collaboratively across multiple participants. In terms of how the data is distributed among the participants of federated learning, federated learning can be roughly categorized into Horizontal Federated Learning (HFL) and Vertical Federated Learning (VFL). HFL is centered on collaborative model training among the devices possessing the same feature set but with non-overlapping samples (McMahan et al., 2017; Karimireddy et al., 2020; Li et al., 2020; 2021; Marfoq et al., 2022; Mishchenko et al., 2019). In contrast, VFL involves collaboration among participants with non-overlapping feature sets but on the same sample (Vepakomma et al., 2018; Yang et al., 2019; Liu et al., 2019; Chen et al., 2020; Gu et al., 2020; Zhang et al., 2021c;b;a; Wang et al., 2023; Zhang et al., 2024; Qi et al., 2022).

Research on VFL is facing a wide variety of challenges. The primary concern in VFL lies in privacy and security. To protect the privacy, VFL frameworks commonly employ privacy protection methods such as differential privacy (DP) (Ranbaduge & Ding, 2022; Wei et al., 2020; Zhou et al., 2020; Huang et al., 2015; Zhou et al., 2020; Huang et al., 2015), homomorphic encryption (HE) (Hardy et al., 2017; Liu et al., 2020), secure multiparty computation (SMC)(Fang et al., 2021; Mugunthan et al., 2019; Gu et al., 2020). The second challenge encountered in VFL pertains to communication costs. This is due to the transmission of large intermediate model information through the network during VFL training. To enhance communication efficiency in federated learning, the most common method is to compress the communication of the VFL (Castiglia et al., 2022; Wang et al., 2022; 2023). Besides, another common strategy involves using multiple local updates on each participant to reduce the number of communication rounds (Liu et al., 2019; Fu et al., 2022). However, these methods often assume the client possesses additional information, such as data labels, which may not align with the general VFL framework. The third challenge of VFL involves adapting to a new data distribution efficiently. The distribution of data may change throughout the life cycle of VFL. Offline learning settings require retraining the VFL model when encountering concept drift, which is neither computationally efficient nor communication efficient within the VFL framework. Therefore, applying online learning to the VFL framework is essential (Wang & Xu, 2023). Moreover, online VFL also alleviates storage costs, particularly when clients have limited storage space. In the offline learning paradigm, both the client and the server possess large datasets. Certain types of VFL based on the offline learning paradigm are required to store the intermediate results corresponding to each sample which takes a large storage cost (Chen et al., 2020; Wang et al., 2023; Zhang et al., 2021a). In contrast, online learning obviates the need to store large datasets. Participants receive

one sample from the environment at a time, making it suitable for VFL scenarios with low-capacity participants, such as sensor networks (Wang & Xu, 2023; Liu et al., 2022).

## E.2 Online Learning

Online learning is a paradigm wherein models are continually updated as fresh data becomes accessible, as opposed to being trained on a static dataset in a batch mode. In the well-established online convex optimization, the primary objective is to minimize regret (Hoi et al., 2018; Hazan et al., 2016), which is the difference between the cumulative loss of the player and the optimal solution. The most straightforward approach to online convex optimization involves directly optimizing regret, known as *follow the leader (FTL)* (Hazan et al., 2016). However, FTL can be unstable which motivates the need to stabilize the training through regularization. Therefore the idea of *follow the regularized leader (FTRL)* (Shalev-Shwartz & Singer, 2007; Abernethy et al., 2009; Hazan & Kale, 2010; McMahan et al., 2013) was proposed to address this problem. In FTL and FTRL, the learner is required to store all previously seen samples, which is inefficient in terms of memory and computation. Another prevalent algorithm for online convex optimization is online gradient descent (OGD) (Zinkevich, 2003). This iterative optimization algorithm updates the model using the gradient on the incoming data point. In theory, it achieves a sublinear regret of $O(\sqrt{T})$. Hazan et al. (2007) further introduced the adaptive OGD algorithm, which offers intermediate regret rates between $O(\sqrt{T})$ and $O(\log(T))$.

While traditional online learning predominantly addresses convex cases with shallow models, recent research has shifted its focus towards the non-convex case of online learning. Hazan et al. (2017) introduced the concept of *local regret* as a surrogate for regret analysis in non-convex online learning. Unlike the regret utilized in online convex optimization, the fundamental idea is to confine the regret within a sliding window, rendering it "local". Aydore et al. (2019) further explores the concept of local regret, introducing *dynamic local regret* to address concept drift in the data stream. This approach incorporates an exponential average over the sliding window of local regret. Additionally, they utilize past gradients for the window, enhancing computational efficiency compared to the static local regret proposed by Hazan et al. (2017). Gao et al. (2018) present an online normalized gradient descent algorithm for scenarios with gradient information available and a bandit online normalized gradient descent algorithm when only loss function values are accessible. They demonstrate achieving a regret bound of $O(\sqrt{T + V_T T})$. Sahoo et al. (2018) propose online deep learning to tackle online learning within the deep learning paradigm. Agarwal et al. (2019) addresses online learning in a non-convex setting by leveraging an offline optimization oracle. Their study demonstrates that by enhancing the oracle model, online and statistical learning models achieve computational equivalence. Suggala & Netrapalli (2020) demonstrate achieving an $O(\sqrt{T})$ rate using the Follow the Perturbed Leader (FTPL) algorithm. Héliou et al. (2020) address online learning with non-convex losses, introducing a mixed-strategy learning policy based on dual averaging under the assumption of inexact model feedback for the loss function.

To facilitate a clear comparison, we present Table 6 which summarizes the regret bounds of the most relevant works, including both standalone online learning algorithms and the online VFL. Specifically, compared to the theoretical results of standalone DLR (Aydore et al., 2019), the additional constant term $(2W\mathbf{G})$ arises from the missing gradient elements of passive clients due to the dynamic partial activation of clients in the event-driven online VFL framework.

Table 6: Comparison of the regret bound

| Method | Online Convex Learning | Online Non-Convex Learning |
|---|---|---|
| **Standalone** | | |
| OGD (Hazan et al., 2016) | $O(\sqrt{T})$ | - |
| SLR (Hazan et al., 2017) | - | $SLR_w(T) \leq \frac{T}{w}(8\beta M + \sigma^2)$ |
| DLR (Aydore et al., 2019) | - | $DLR_w(T) \leq \frac{T}{W}(8\beta M + \sigma^2)$ |
| **Online VFL** | | |
| Online VFL (Wang & Xu, 2023) | $O(\sqrt{T})$ | - |
| Event-Driven Online VFL (ours) | $O(\sqrt{T})$ | $DLR_w(T) \leq \frac{T}{W} \cdot \frac{p_{max}}{p_{min}} \cdot \left( \frac{8\beta M}{p_{max}} + 2\sigma^2 + 2W\mathbf{G} \right)$ |

