# OpenReview forum: "Event-Driven Online Vertical Federated Learning"
_ICLR.cc/2025/Conference — ICLR 2025 Poster_

### Official Review · Reviewer_MnZ6 · 2024-10-20

**Soundness:** 4
**Presentation:** 4
**Contribution:** 3
**Rating:** 8
**Confidence:** 4

**Summary:**

This work tackles a practical problem with online and vertical federated learning where clients do not receive the streaming data synchronously. In other words, the features of a data sample are assigned to a few of the participating clients in an event-driven manner. The work proves its empirical superiority against traditional online regret minimization and static local regret baselines. A sublinear regret bound for non-convex models is also derived.

**Strengths:**

1. The writing is very clear.
2. The work is novel; addressing the practical setting of streaming data for vertical federated learning (for non-convex models) is timely and relevant.
3. The proposed solution is elegant, and the authors have done a good job at rationalizing each component of their methodology.

**Weaknesses:**

1. I did not fully get the significance of passive clients. Are passive clients also assigned unique features of a data sample? If the features assigned to the passive clients are not unique, then why do we need the derived embeddings from the passive clients?

2. I acknowledge that the work is based on a cross-silo or vertical FL setting, where a small quantity of clients is the norm. However, experiments with 4 clients and a 3-layer model still make me question how scalable this work is.

**Questions:**

1. What are the differences between the use of dynamic local regret proposed in the Aydore paper cited and your work? Is it that in your case, half of the model is on server and the other half is at clients? What specific challenges did you encounter in adapting the said dynamic local regret to your framework?

2. For eq. 1, are embeddings from each client concatenated?

3. Referring to lines 162-163, is $W=M$?

4. I acknowledge that the discussion related to the following question is provided in "Limitations." However, I still want to clarify this point: Are all features of a data sample "covered" (trained on) by all the activated clients?

5. Do we always need to divide the features of a data sample among all the participating clients? Can there be overlapping features across clients? Can there be multiple clients activated for the same feature? Under the "Event" experimental setting, do all four clients participate each round? If not, what happens to the unassigned features?

6. Do all activated clients receive the assigned features at the same time/synchronously?

7. Referring to line 278, is capital $F$ in $\nabla F^t (w_o, \mathbf{w})$ a typo?

8. I am curious to know if an even distribution of features across the clients is a practical assumption (with reference to line 322).

9. Is a batch of embeddings from a client sent back to the server? Or the embeddings are also sent in a streaming setup?

10. How do you define a round for the streaming/online FL?

---

> ### Author Response · Authors · 2024-11-20
> **Response to Reviewer MnZ6 - Part 1**
>
> Thank you very much for the high praise and support of our work. We are deeply honored and motivated by your kind words. We reply to the weaknesses and questions.
>
> > *W1: I did not fully get the significance of passive clients. Are passive clients also assigned unique features of a data sample? If the features assigned to the passive clients are not unique, then why do we need the derived embeddings from the passive clients?*
>
> In VFL, the features held by different clients are **unique/non-overlapping** by definition, reflecting real-world application scenarios. For example, in a banking VFL setting, Bank A holds a transaction record of the Bank A account for a customer, while Bank B holds the transaction records of the Bank B account for the same customer. These records are non-overlapping by definition as each bank only accesses its own data. Refer to figure 1 in [Wei et al., (2022)](https://arxiv.org/pdf/2202.04309) for a vivid illustration.
>
> Passive clients are essential because the server model relies on embeddings from all clients to ensure a **complete view of the input** for the learning process (Eq. 1). As a theoretical research, including passive clients, guarantees that the server operates with a **full set of correct inputs**, enabling accurate model updates and analysis.
>
> In practice, the participation of passive clients can be omitted by using default embeddings to approximate the missing ones. However, this approach significantly increases the complexity of theoretical analysis due to the resulting **incomplete view problem**, as discussed in Section 6. This problem would be a critical direction for future research.
>
>
> > *W2: I acknowledge that the work is based on a cross-silo or vertical FL setting, where a small quantity of clients is the norm. However, experiments with 4 clients and a 3-layer model still make me question how scalable this work is.*
>
> Thank you for your suggestion. In response, we extended our main experiments in Section 5.2 to include scenarios with **8 and 16 clients**. The corresponding results are available via this anonymous [\[clickable link\]](https://anonymous.4open.science/r/EventDrivenOnlineVFL_ICLR_Disucssion-2748/iMNIST/iMNIST_IID_client8_subfig.png).
> These experimental results will be added in the appendix. The conclusion remains consistent with our previous findings: OGD is less stable under partial client activation, while DLR converges more rapidly than SLR. This consistency supports the robustness and scalability of our proposed framework.
>
>
> > *Q1: (1) What are the differences between the use of dynamic local regret proposed in the Aydore paper cited and your work? (2) Is it that in your case, half of the model is on server and the other half is at clients? (3) What specific challenges did you encounter in adapting the said dynamic local regret to your framework?*
>
> (1) The primary difference lies in the **design of the distributed DLR buffer**. DLR (Aydore et al. 2019) was originally designed for standalone models. However, in the VFL setting, the model is no longer standalone, requiring a distributed buffer design for the server and the clients. In particular, designing the client's buffer is the most challenging part due to the uncertainty of whether a client will be activated by the event in a given round. **The clients' buffer must function effectively in both scenarios**. Following this principle, we developed the client's procedure (Algorithm 1) and buffer structure (Eq. 5).
>
> (2) Yes, the server holds the upstream model, represented as $f(w_0, \cdots; y)$ in Eq. 1, while the clients hold the downstream models, represented as $h_m(w_m; x)$, $m \in [M]$ in Eq. 1.
>
> (3) As mentioned earlier in (1), designing the clients' DLR buffer is the most challenging part. Additionally, it also poses challenges for regret analysis due to the **asymmetry between the server's and clients' buffers during runtime** (the server is always active, while client activation is uncertain). If the server and clients had symmetric updates in VFL, the convergence analysis could be simplified by treating the entire VFL as a single global model (Liu et al., 2019; Castiglia et al., 2022). However, when the updates of the server and clients differ, as in event-driven online VFL, the regret analysis becomes significantly more complex.
>
> > *Q2: For eq. 1, are embeddings from each client concatenated?*
>
> Our work is based on a general VFL framework. Eq. 1 specifies only that the embeddings $h_1, ... h_M $ are provided as inputs to the server, without restricting the operations used. These could include concatenation, summation, or any other methods. In our experiments, we use concatenation.

---

> ### Author Response · Authors · 2024-11-20
> **Response to Reviewer MnZ6 - Part 2**
>
> > *Q3: Referring to lines 162-163, is W=M?*
>
> Thank you for pointing this out. The $M$ here is a typo and should be corrected to $W$.
>
> > *Q4: I acknowledge that the discussion related to the following question is provided in "Limitations." However, I still want to clarify this point: Are all features of a data sample "covered" (trained on) by all the activated clients?*
>
> No, the activated clients do not cover all the features of an incoming data sample. The complete feature set of a data sample is distributed across all clients, including both activated and passive clients.
>
> > *Q5: (1) Do we always need to divide the features of a data sample among all the participating clients? (2) Can there be overlapping features across clients? (3) Can there be multiple clients activated for the same feature? (4) Under the "Event" experimental setting, do all four clients participate in each round? If not, what happens to the unassigned features?*
>
> (1) In real-world applications, data are generated by the participants of VFL, meaning the features are distributed across clients from the start. In VFL research experiments, the standardized procedure is manually dividing the features of a single dataset (all references in lines #32-#34).
>
> (2) In the classical VFL setting, features are **non-overlapping** across clients. The sample ID is assumed to be shared but is not used for learning purposes.
>
> (3) As the features are non-overlapping across clients, only one client can be activated by one given feature on its side (if this feature is the indicator).
>
> (4) All clients participate in each round, with both activated and passive clients processing their respective features. There is no unassigned feature.
>
> > *Q6: Do all activated clients receive the assigned features at the same time/synchronously?*
>
> As a theoretical work, we model this process by assuming that activated clients detect the event synchronously at the same time step $t$. However, in real-world applications, ideal synchronicity is rarely achievable; therefore, activated clients within a given "time window" may be treated as a single event. For example, in a 5-second time window, multiple sensors may be activated upon detecting an event.
>
> > *Q7: Referring to line 278, is capital $F$ in $\nabla F^t(\cdot)$ a typo?*
>
> Thank you for pointing that out. Yes, it should be corrected to $\nabla f(\cdot)$.
>
> > *Q8: I am curious to know if an even distribution of features across the clients is a practical assumption (with reference to line 322).*
>
> We acknowledge that in some real-world applications, feature sizes may vary across clients. However, applying an even distribution of features among clients is a standardized experimental procedure in VFL experiments (all references in lines 32-34). This approach ensures a standardized data processing method, facilitates comparison with future research, and simplifies the framework's implementation.
>
> > *Q9: Is a batch of embeddings from a client sent back to the server? Or the embeddings are also sent in a streaming setup?*
>
> The embeddings are transmitted in a streaming setup, i.e. in each round, the client receives its portion of the data, processes it using its model $h_m(\cdot)$, and sends the resulting embedding to the server. Our research follows the classic online learning setting, where each time step involves processing a single data point $x_t$, and clients do not aggregate multiple data points into batches.
>
> > *Q10: How do you define a round for the streaming/online FL?*
>
> One round of online VFL consists of the following steps: receiving a single data point, making a prediction, obtaining feedback (label) from the environment, and updating the model.
>
> In our proposed framework, Figure 1 illustrates the sequence of operations in one round of the event-driven online VFL process: Event occurrence and client activation -> The activated clients send embeddings to the server -> The server queries the passive clients -> The server replies to the activated clients -> Server updates Eq. 4 -> Activated client update, Eq. 5.

---

> > ### Comment · Reviewer_MnZ6 · 2024-11-22
> > **Thank you for the answers**
> >
> > I would like to keep my score. Thank you to the authors for providing further details and clarifications.

---

> > > ### Author Response · Authors · 2024-11-24
> > > **Thank you!**
> > >
> > > Thank you again for your kind words and for taking the time to review our paper! Your high score and support are incredibly encouraging to us!
> > >
> > > We are continuously improving the manuscript, particularly by adding the experimental results shared via the anonymous link during the discussion period.

---

### Official Review · Reviewer_x1ux · 2024-10-29

**Soundness:** 3
**Presentation:** 3
**Contribution:** 3
**Rating:** 6
**Confidence:** 4

**Summary:**

This paper proposes an online vertical federated learning framework,  based on the mechanism of exponential weighted sliding-window averaging. The dynamic local regret analysis by Aydore et al. is employed and extended to analyze this framework. Typically, only a subset of clients were activated during each event, and meanwhile the remaining clients can be reached out for passive collaboration during the learning process. Dynamic local regret (DLR) can be derived from the analytical framework, under some reasonable assumptions. Finally, experiments are conducted under various settings to verify the theoretical results.

**Strengths:**

S1. A reasonable setting of VFL is considered.

S2. The proposed solution is general (for a large class of learning algorithms) and easy to be implemented.

S3. Some sound theoretical results are derived under reasonable assumptions.

S4. Experiments are conducted under various setting.

**Weaknesses:**

W1. The theoretical contribution is incremental.

W2. More discussion about the implication of theoretical conclusions about DLR in practice is needed.

**Questions:**

Q1. Is $x$ missing from $f^t(w_0, {\bf w},)$ in (1)? Should it be something like $f^t(w_0, {\bf w}, x; y)$?

Q2. What is the biggest challenge when extending the results about DLR in Aydore et al. 2019 to the setting of this paper?

Q3. Is it possible to extend the proposed framework for HFL? If not, why? If yes, how does it compare with previous works on online HFL?

---

> ### Author Response · Authors · 2024-11-20
> **Response to Reviewer x1ux**
>
> Thank you for your insightful comments and for recognizing the contribution of our work. Your feedback has been invaluable in enhancing the quality and clarity of our manuscript. We reply to the weaknesses and questions.
>
> > *W1: The theoretical contribution is incremental.*
>
> The theoretical contribution of our work stems from **identifying the research problem** and **addressing the theoretical challenges in designing event-driven online VFL**.
>
> **Identify the research problem:** In online VFL, the clients receive non-overlapping features of data from the environment. Previous online-VFL research naively assumes that all client receives a synchronous data stream. In that naive case, the online-VFL problem reduces to the standalone online learning problem, which is lack of novelty and challenges. However, upon observing real-world applications of VFL within banking institutions, companies, and sensor networks, we found that this naive assumption rarely holds true; it is uncommon for all clients to receive the features simultaneously. Therefore, we **identified the challenges in online VFL that have been overlooked in previous research** and proposed a framework that is **more applicable to real-world scenarios**.
>
> **Theoretical challenges and innovation:** Adapting DLR to VFL is not straightforward in both algorithm design and theoretical analysis. DLR (Aydore et al., 2019), was originally designed for standalone models. However, in the VFL setting, the model is no longer standalone, requiring a **distributed buffer design** for the server and the clients. In particular, designing the client's buffer is the most challenging part due to the **uncertainty of whether a client will be activated or not** in a given round in the event-driven online VFL. The buffer must function effectively in both scenarios. Following this principle, we carefully developed the client's procedure (Algorithm 1) and buffer structure (Eq. 5).
>
> Moreover, the uncertainty of client activation introduces additional **asymmetry between the server's and clients' buffers during runtime** (the server is always active, while client activation is uncertain), posing significant challenges for regret analysis. In some classical VFL research (Liu et al., 2019; Castiglia et al., 2022), where the server and clients had symmetric updates, convergence analysis can be simplified by treating the entire VFL system as a single global model. However, when the updates of the server and clients differ, the convergence analysis of VFL becomes considerably more complex.
>
> > *W2: More discussion about the implication of theoretical conclusions about DLR in practice is needed.*
>
> Thank you for the suggestion. We will add more illustrations in the Theorem section.
>
> First, the primary implication of the theorem is that it guarantees the regret, as defined under the DLR framework, grows sublinearly with time $T$. This demonstrates the effectiveness of the proposed algorithm in addressing the event-driven online VFL problem in the non-convex setting.
>
> Second, compared to the theoretical results of standalone DLR (Aydore et al., 2019), the additional constant term arises from the missing gradient elements of passive clients due to the dynamic partial activation of clients in the event-driven online VFL framework (corollary 2).
>
>
> > *Q1: Is $x$ missing from $f_t(w_0,w)$ in (1)? Should it be something like $f_t(w_0,w,x;y)$?*
>
> Yes, $f^t(w_0, w)$ is an abbreviation of $f^t(w_0, w, x^t; y^t)$. We will include this clarification for better illustration.
>
> > *Q2: What is the biggest challenge when extending the results about DLR in Aydore et al. 2019 to the setting of this paper?*
>
> *Refer to the response to W1*: The primary challenges include the **design of the distributed DLR buffer**, particularly addressing the **uncertainty of client activation**, and the **asymmetry between server and client updates**, which introduces significant complexity in regret analysis.
>
>
> > *Q3: Is it possible to extend the proposed framework for HFL? If not, why? If yes, how does it compare with previous works on online HFL?*
>
> Applying online learning to HFL and VFL is totally different. Applying online learning to **HFL** is much more straightforward than to VFL. In HFL, each client receives a **complete data stream** $(x^t, y^t)$ and maintains **a copy of the global model**. With these resources in place, the clients in HFL can easily apply **any standalone online learning approach locally**.
> Moreover, **handling a subset of activated clients is inherently part of the HFL framework** (McMahan et. al. 2017).
>
> In contrast, **online VFL** is a totally different case, where no research has formed on these fundamental topics in distinct data streams or the partial client activation.

---

> ### Author Response · Authors · 2024-11-25
> **Follow-Up on Reviewer Feedback and Submission Updates**
>
> **Thank you very much for taking the time to review our paper. We sincerely appreciate your valuable feedback and for recognizing the contribution of our work. Does our response address your concerns?**
>
> Additionally, we have updated the submission, enhancing the clarity of Equation 1 based on our discussion in Q1.

---

### Official Review · Reviewer_qrpn · 2024-11-03

**Soundness:** 3
**Presentation:** 3
**Contribution:** 3
**Rating:** 8
**Confidence:** 4

**Summary:**

This paper discusses vertical federated learning (VFL) in an online setting when all clients receive a synchronous data stream. Instead of reviewing the updates of a model from a client-focus aspect, this paper proposes to review this problem through an event-driven online VFL framework. That is, only a subset of clients were activated during each event, while the remaining clients passively collaborated in the learning process. As this will lead to the non-convex optimisation problem, a dynamic local regret approach is adapted to handle online learning in non-convex cases and non-stationary environments.

**Strengths:**

The idea is novel, and the presentation is clear.

**Weaknesses:**

The solution is simple. As there are not many available baselines for direct comparison, the experimental results can only demonstrate the proposed solution is a feasible plan. Therefore, although the limitations of the proposed framework have been discussed at the end, it lacks the support from experiments to form a deep understanding.

**Questions:**

1. I might miss the result of SLR for SUSY and HIGGS datasets. Where can I find that?
2. Here, each of the images in iMINST has been divided into 4 segments. However, it didn't specify if segments are overlapped with each other.
3. If the segments can have overlaps, I would like to know the results of having more than 4 clients, especially how the performance might be influenced when overlaps increase.
4. If the segments do not have overlaps, please either explain why or add the results for an overlapped case.
5. Is only one client activated each time? If yes, can multiple clients be activated at the same time?

---

> ### Author Response · Authors · 2024-11-20
> **Response to Reviewer qrpn - Part 1**
>
> We sincerely thank you for your insightful comments and constructive criticisms. Your feedback has been invaluable in improving the quality and clarity of our manuscript. Below, we address the weaknesses and respond to your questions.
>
> > *W1: The solution is simple. As there are not many available baselines for direct comparison, the experimental results can only demonstrate the proposed solution is a feasible plan. Therefore, although the limitations of the proposed framework have been discussed at the end, it lacks the support from experiments to form a deep understanding.*
>
> While the proposed method looks "simple", it encounters **significant challenges in both algorithm design and theoretical analysis**.
>
> **Challenges of distributed DLR buffer design:** DLR (Aydore et al., 2019), was originally designed for standalone models. However, in the VFL setting, the model is no longer standalone, requiring a separate buffer design for the server and the clients (section 3.2). In particular, designing the client's buffer is the most challenging part due to the **uncertainty of whether a client will be activated or not in a given round**. The buffer must function effectively in both scenarios. Following this principle, we developed the client's procedure (Algorithm 1) and buffer structure (Eq. 5).
>
> Moreover, the uncertainty of client activation introduces **asymmetry between the server's and clients' buffers during runtime** (the server model is always activated, while the client's activation is uncertain), posing significant challenges for regret analysis. In some classical VFL research (Liu et al., 2019; Castiglia et al., 2022), where the server and clients had symmetric updates, convergence analysis can be simplified by treating the entire VFL system as a single global model. However, when the updates of the server and clients differ, as in event-driven online VFL, the convergence analysis of VFL becomes considerably more complex.
>
>
> **Regarding the baselines:** The primary reason for the lack of research baseline on online-VFL is that naively applying online learning to VFL assumes that all client receives a synchronous data stream. In that naive case, the online-VFL problem **reduces to the standalone online learning problem**, which is lack of novelty and challenges. As a result, research in online VFL has faced obstacles, leaving no existing works available for direct comparison.
>
> However, upon observing real-world applications of VFL within banking institutions, companies, and sensor networks, we found that this naive assumption rarely holds true; it is uncommon for all clients to receive the features simultaneously. Therefore, we **identified the challenges inherent to the nature of VFL that have been overlooked in previous research** and proposed a **novel framework that is more applicable to real-world scenarios**. Through the exploration of event-driven mechanisms, we open up new possibilities for data streaming processing across distributed nodes within the VFL framework.
>
> Moreover, In our experimental section, we included **as many relevant baselines as possible**. Specifically, we adapted the online VFL framework from Wang & Xu (2023), which uses OGD for optimization. Besides, we designed our own adaptation of Static Local Regret (SLR) (Hazan et al., 2017) for the event-driven online VFL setting (Appendix C.3) as additional baselines for online non-convex learning. These serve as competitive baselines for comparison with our proposed DLR framework. Furthermore, for each optimization framework (OGD, SLR, DLR), we implemented both the naive "Full" framework and the partial activation framework (Random & Event), ensuring a comprehensive evaluation across different settings.

---

> ### Author Response · Authors · 2024-11-20
> **Response to Reviewer qrpn - Part 2**
>
> > *Q1: I might miss the result of SLR for SUSY and HIGGS datasets. Where can I find that?*
>
> That is our miss. We did not add the SLR line to make the left and right figures look neat and easier for comparison in Appendix D.2. We provide these anonymous clickable links for the [\[SUSY\]](https://anonymous.4open.science/r/EventDrivenOnlineVFL_ICLR_Disucssion-2748/SUSY/SUSY_nonIID_subfig.png) and [\[HIGGS\]](https://anonymous.4open.science/r/EventDrivenOnlineVFL_ICLR_Disucssion-2748/HIGGS/HIGGS_nonIID_subfig.png) experiment with the SLR lines. These figures will replace the original ones.
>
>
> > *Q2, Q3, Q4: Here, each of the images in iMINST has been divided into 4 segments. However, it didn't specify if segments are overlapped with each other. If the segments can have overlaps, I would like to know the results of having more than 4 clients, especially how the performance might be influenced when overlaps increase. If the segments do not have overlaps, please either explain why or add the results for an overlapped case.*
>
> In VFL, the features (segments) held by different clients are **non-overlapping** by definition, reflecting the nature of real-world application scenarios. For example, in a banking VFL setting, Bank A holds a transaction record of the Bank A account for a customer, while Bank B holds the transaction records of the Bank B account for the same customer. These records are non-overlapping by definition as each bank only accesses its own data. This applies similarly to other VFL application scenarios. Refer to figure 1 in [Wei et al., (2022)](https://arxiv.org/pdf/2202.04309) for a vivid illustration.
>
> > *Q5: Is only one client activated each time? If yes, can multiple clients be activated at the same time.*
>
> No, multiple clients are activated in each round, as mentioned in the introduction (lines #83-#84) and further detailed in Section 3.3.

---

> > ### Author Response · Authors · 2024-11-25
> > **Follow-Up on Reviewer Feedback and Submission Updates**
> >
> > **Thank you very much for taking the time to review our paper. We sincerely appreciate your valuable feedback and have carefully addressed the concerns in our previous response. Does our response address your concerns?**
> >
> > Besides, we have updated the submission. The experimental results discussed in **Q1** (SLR line for SUSY and HIGGS) have been updated in **Appendix D.2**.

---

> > ### Comment · Reviewer_qrpn · 2024-11-25
> >
> > Thanks for your answer.
> >
> > In lines #83-#84, it does say "a subset of the clients are activated". My question here is to ask how this multiple activation is implemented during your experiment. As you have four clients only, a subset can be 1,2,3 client(s) activated. If it has to be multiple clients activated in each round, we have 2 or 3 clients activated. 3 clients are activated means that only 1 client is in the other group, which seems breaking the law of "multiple". Therefore, I am wondering if this point does matter. Basically, it is to ask if you have restricts such as the number of activated clients must be more than 1 and meanwhile the  number of inactivated clients must be more than 1

---

> ### Author Response · Authors · 2024-11-25
> **Response to Reviewer qrpn's Question on Client Subset Activation**
>
> Thank you for your question.
>
> We apologize for the confusion caused by our earlier response to Q5. A more accurate reply to Q5 should be: *"multiple clients **can be** activated in each round"*, rather than *"multiple clients **are activated** in each round"*. This phrasing better emphasizes the capability of activating more than one client, while not excluding the possibility of 1 or 0 clients being activated. Throughout the paper, we only use the term **"subset"**, which means that a "subset of clients activated by the event" can include **0**, 1, 2 ..., or **M** clients, covering all possibilities from an **empty set** to the **complete set**.
>
> In the experiments, the "Random" framework activates each client with a probability $p$ (line #340), while the "Event" framework activates a client if the average of its input features exceeds a threshold $\Gamma $ (line #343). We do not impose a strict requirement on "multiple clients" (more than 1) in our setup. Instead, the aforementioned design allows for scenarios with a single active client or even no activated clients (e.g., in some application scenarios, the event may occur on the server side, resulting in 0 activated clients). This flexibility creates a more general framework that can accommodate a wide range of application scenarios.
>
> We hope this clarifies your concerns, and we appreciate your attention to these details. Please let us know if there are further points we can address.

---

> > ### Comment · Reviewer_qrpn · 2024-11-26
> >
> > Thanks.
> >
> > If that is the case, is that possible to know how many clients are activated during the experiment? It would be more clear on the contribution of this paper if knowing whether it is always 0, or 4 or at a fixed number.

---

> ### Author Response · Authors · 2024-11-27
> **Response to Reviewer qrpn's Question on Client Activation Monitoring and Statistical Analysis**
>
> Thank you for your suggestion! It provides valuable insight into the partial client activation mechanism of our framework.
>
> Actually, we did implement a function to **monitor and record the frequency of activation for each client** throughout the training process. This ensures that extreme scenarios, such as "always 0" or "always full" activation, are avoided in the main experiments.
>
> To provide more insight, we present the following table summarizing the **activation statistics** for each client under different event-driven framework settings in Section 5.4. For reference, we additionally included the extreme cases for $\Gamma = +\infty$ (always 0) and $\Gamma = -\infty$ (always full). This table will also be added to the **appendix**.
>
> Table 1: Frequency of Activation for Each Client Under Different Settings
>
> | Setting       | Client 1        | Client 2        | Client 3        | Client 4        |
> |-----------------|-----------------|-----------------|-----------------|-----------------|
> | **Random**      |                 |                 |                 |                 |
> | $p = 0.25$      | 0.2497465       | 0.2504725       | 0.2502765       | 0.250142        |
> | $p = 0.5$       | 0.4995065       | 0.499811        | 0.500266        | 0.500179        |
> | $p = 0.75$      | 0.7496225       | 0.750214        | 0.7501545       | 0.750257        |
> | $p = 1.0 $      | 1.0             | 1.0             | 1.0             | 1.0             |
> | **Event**       |                 |                 |                 |                 |
> | $\Gamma = +\infty$  (+100) | 0          | 0               | 0               | 0               |
> | $\Gamma = 0.6$  | 0.0000055         | 0.066729        | 0.101827        | 0.0002035       |
> | $\Gamma = 0.2$  | 0.0063455       | 0.4611815       | 0.456317        | 0.038649        |
> | $\Gamma = -0.2$ | 0.355166        | 0.982896        | 0.9852155       | 0.596582        |
> | $\Gamma = -\infty$  (-100) | 1.0        | 1.0             | 1.0             | 1.0             |
>
> **Note**: The values in the table represent the *activation frequency for each client*, calculated as the ratio of *the number of iterations in which the client was activated* to *the total number of iterations*.

---

> > ### Comment · Reviewer_qrpn · 2024-11-27
> >
> > Thanks. All clear.

---

> > > ### Author Response · Authors · 2024-11-27
> > > **Thank you!**
> > >
> > > We are truly grateful for your thoughtful and in-depth discussions! Your support means so much to us!

---

### Official Review · Reviewer_H7dv · 2024-11-04

**Soundness:** 3
**Presentation:** 3
**Contribution:** 2
**Rating:** 6
**Confidence:** 4

**Summary:**

This paper focuses on online vertical federated learning and proposes an innovative event-driven framework. The main contributions are particularly noteworthy: (1) The authors make a novel observation that in real-world scenarios, clients in vertical federated learning are unlikely to receive different features of the same sample synchronously. This perspective is novelty and addresses a significant gap in current research. (2) The authors then develop an event-driven online vertical federated learning framework. A particularly valuable contribution is the incorporation of Dynamic Local Regret into this framework to handle challenges arising from non-convex models in non-stationary environments. (3) This framework effectively bridges the gap between theoretical VFL models and practical applications, addressing real-world challenges that have been overlooked in previous research.

**Strengths:**

1.	The paper makes an insightful observation about asynchronous data reception in VFL - a practical issue that has been surprisingly overlooked in previous research but significantly impacts real-world applications.
2.	The integration of Dynamic Local Regret into VFL shows technical sophistication, offering an elegant solution for non-convex and non-stationary scenarios that extends beyond traditional convex-only approaches.
3.	The theoretical analysis is rigorous, with well-constructed proofs of regret bounds that provide solid mathematical foundation for the event-driven framework.
4.	The practical benefits are clear - by activating only relevant clients, the approach naturally reduces communication and computation overhead, making it more feasible for real-world deployment.

**Weaknesses:**

1.	Unclear innovation contribution

(a) The paper would benefit from a clearer discussion of the specific challenges encountered when adapting event-driven client participation to VFL, along with the corresponding design considerations and solutions proposed to address these challenges.

(b) Similarly, the paper could better elucidate the specific technical challenges encountered in DLR integration and more clearly demonstrate the novel solutions developed to overcome them.

2.	Lack of comparative analysis

The comparative analysis could be expanded. The paper does not discuss several relevant works in online VFL, such as "Online Vertical Federated Learning for Cooperative Spectrum Sensing, Wang et al. " and "Vertical Semi-Federated Learning for Efficient Online Advertising, Li et al.". Including comparisons with these works would help better position this paper's contributions within the existing works.

3.	Lack of comparative analysis

The experimental evaluation would benefit from including standard VFL baselines, such as Local Model and Vanilla VFL in “Fedcvt: Semi-supervised vertical federated learning with cross-view training, Kang et al.”, “VERTICAL FEDERATED LEARNING HYBRID LOCAL PRE-TRAINING, Li et al.”, to provide a more comprehensive comparison of the proposed approach's performance.

4.	Online varying numbers of active clients

Given that the paper focuses on event-driven client participation, and changing event, network disconnections, or other factors, the experimental section would benefit from exploring scenarios with varying numbers of active clients.

**Questions:**

1.	Could this paper include a convergence comparisons with existing approaches? The current convergence analysis lacks comparative results that would help demonstrate the relative convergence performance and efficiency of the proposed method against other existing frameworks.
2.	I am curious about how this paper address privacy in this framework. Could you demonstrate on the acceptable privacy protection mechanisms and safeguards implemented in the proposed approach?
3.	How does the framework's performance scale with different total numbers of clients? It would be valuable to evaluate whether the proposed approach maintains its effectiveness as the number of clients increases or decreases.

---

> ### Author Response · Authors · 2024-11-20
> **Response to Reviewer H7dv - Part 1**
>
> We sincerely thank you for your insightful comments and constructive criticisms. Your feedback has been invaluable in improving the quality and clarity of our manuscript. Below, we address the weaknesses and respond to your questions.
>
> > *W1: Unclear innovation contribution on:
> **(a)** The paper would benefit from a clearer discussion of the specific challenges encountered when adapting event-driven client participation to VFL, along with the corresponding design considerations and solutions proposed to address these challenges.
> **(b)** Similarly, the paper could better elucidate the specific technical challenges encountered in DLR integration and more clearly demonstrate the novel solutions developed to overcome them.*
>
> (a) Challenges of event-driven online VFL:
>
> 1. **Identify the research problem:** In online VFL, the clients receive non-overlapping features of data from the environment. Previous online-VFL research naively assume that all client receives a synchronous data stream. In that naive case, the online-VFL problem **reduces to the standalone online learning problem**, which is lack of novelty and challenges. However, upon observing real-world applications of VFL within banking institutions, companies, and sensor networks, we found that this naive assumption rarely holds true; it is uncommon for all clients to receive the features simultaneously. Therefore, we **identified the challenges in online VFL that has been overlooked in previous research** and proposed a framework that is **more applicable to real-world scenario**.
>
> 2. **Uncertainty of partial client activation:** In event-driven online VFL, the subset of clients activated by the event changes dynamically in each round, introducing uncertainty in client activation. This uncertainty in the client's activation introduces significant challenges for algorithm design and theoretical analysis, as discussed below in (b).
>
> (b) Challenges of DLR integration:
>
> 1. **Distributed DLR buffer design:**  DLR (Aydore et al. 2019), was originally designed for standalone models. However, in the VFL setting, the model is no longer standalone, requiring a distributed buffer design for the server and the clients (section 3.2). In particular, designing the client's buffer is the most challenging part due to the uncertainty of whether a client will be activated or not in a given round. The buffer must function effectively in both scenarios. Following this principle, we developed the client's procedure (Algorithm 1) and buffer structure (Eq. 5).
>
> 2. **Asymmetry in server and client's update:** The asymmetry between the server's and clients' buffers during runtime introduces significant challenges in regret analysis (The server model is always activated, while the client's activation is uncertain). If the server and clients had symmetric updates in VFL, the convergence analysis could be simplified by treating the entire VFL as a single global model, parameterized by $\Theta$ (Liu et al., 2019; Castiglia et al., 2022). However, when the updates of the server and clients differ, as in event-driven online VFL, the regret analysis becomes significantly more complex.
>
> > *W2: The comparative analysis could be expanded. The paper does not discuss several relevant works in online VFL, such as "Online Vertical Federated Learning for Cooperative Spectrum Sensing, Wang et al." and "Vertical Semi-Federated Learning for Efficient Online Advertising, Li et al.". Including comparisons with these works would help better position this paper's contributions within the existing works.*
>
> The first work has been cited in our related work (line #129, "Wang & Xu, 2023") and has been served as an **important baseline in our experiment section** which we have adapted it as the "OGD-Full" baseline (line #346).
> Moreover, while Wang et al. (2023) focus on synchronous streaming data and online convex learning, we further extended their framework to an event-driven setting (OGD-Random/Event), which serves as additional baselines in our study.
>
> The second work also cited as an important trial addressing the non-overlapping sample problem in VFL (line #513: Li et al. 2022). However, its setting is **offline VFL**, relying on a static dataset to perform self-supervised learning. This approach is fundamentally incompatible with the online learning setting in our research.
> It is worth clarifying that the term **"online advertising"** in their title refers to **"internet-based advertising"** as the application scenario, rather than to "online machine learning".
>
> Regarding the comparative study, we have included related work in the most relevant fields: **online HFL and online VFL** in section 2, with further reviews on **VFL** and **online learning** provided in Appendices E.1 and E.2, respectively.

---

> ### Author Response · Authors · 2024-11-20
> **Response to Reviewer H7dv - Part 2**
>
> > *W3: The experimental evaluation would benefit from including standard VFL baselines, such as Local Model and Vanilla VFL in “Fedcvt: Semi-supervised vertical federated learning with cross-view training, Kang et al.”, “VERTICAL FEDERATED LEARNING HYBRID LOCAL PRE-TRAINING, Li et al.”, to provide a more comprehensive comparison of the proposed approach's performance.*
>
> Thank you for suggesting additional standard VFL baselines for comparison. Although these baselines (Kang et al., Li et al.) are relevant to general VFL research, they are designed for **offline learning scenarios**. Both works rely on a static, large dataset to perform self-supervised learning or pretraining, which is **incompatible with the online learning setting** of our research. Nonetheless, we acknowledge their contributions as efforts to address the non-overlapping view problem in VFL and will ensure they are appropriately cited.
>
> > *W4: Online varying numbers of active clients: Given that the paper focuses on event-driven client participation, and changing event, network disconnections, or other factors, the experimental section would benefit from exploring scenarios with varying numbers of active clients.*
>
> Thank you for your suggestion. Following your suggestion, we conducted an additional experiment in which a certain number of clients were randomly selected for activation, varying the number of activated clients. The results are available via this anonymous [\[clickable link\]](https://anonymous.4open.science/r/EventDrivenOnlineVFL_ICLR_Disucssion-2748/iMNIST/ablation/iMNIST_ablation_Num_Act_Client_DLR.png). This experiment will be added in the appendix section.
>
> We would like to clarify that the experiment varying "the number of active clients" **has been provided in Section 5.4**. where changes to "client activation probability $p$" and "event activation threshold $\Gamma$" **indirectly affect the number of active clients**. Note that in the event-driven online VFL, the number of activated clients is determined by the scope of the event's impact rather than being directly controlled by the server. Therefore, our experiment in that section is based on varying $p$ and $\Gamma$.
>
> > *Q1: Could this paper include a convergence comparisons with existing approaches?*
>
> Thank you for your suggestion. We have added a table summarizing the regret bounds of the most closely related works, which will be included in Appendix E.2 after the related works in Online Learning.
>
> | Method                  | Online Convex Learning | Online Non-Convex Learning |
> |-------------------------|------------------------|----------------------------|
> | **Standalone**          |                        |                            |
> | OGD (Hazan et al., 2016)| $O(\sqrt{T})$          | -                          |
> | SLR (Hazan et al., 2017)| -          | $E[R_w(T)] \le \frac{T}{w}(8 \beta M + \sigma^2 ) $              |
> | DLR (Aydore et al., 2019)| -          | $DLR_w(T) \le \frac{T}{W}(8 \beta M + \sigma^2)$  |
> | **Online VFL**          |                        |                           |
> | Online VFL (Wang & Xu, 2023)   | $O(\sqrt{T})$   | -                         |
> | Event-Driven Online VFL (ours) | $O(\sqrt{T})$   | $ DLR_w(T) \le \frac{T}{W} \frac{p_{max}}{p_{min}}\cdot (\frac{8 \beta M }{p_{max}}  + 2 \sigma^2 + 2 W \mathbf{G} ) $             |
>
> **Note for above table**: $\beta$ is the Lipschitz constant. In SLR, $w$ refers to the window length. In DLR, $W$ is the normalized parameter as defined by Aydore et al. (2019). In our framework, we replace $L$ with $\beta$, and $l$ with $w$ and reorganize the equation for a clear comparison.
>
>
> > *Q2: I am curious about how this paper address privacy in this framework. Could you demonstrate on the acceptable privacy protection mechanisms and safeguards implemented in the proposed approach?*
>
> We appreciate the reviewer's interest in privacy within our proposed framework. As a general VFL framework, it is compatible with mainstream privacy protection mechanisms such as Differential Privacy (DP), Homomorphic Encryption (HE), and Secure Multiparty Computation (SMC).  E.g., Gaussian noise can be added to gradients (DP), or communication between participants can be encrypted using HE to enhance privacy as needed.
> Besides, the partial activation mechanism in our framework inherently provides some privacy protection by limiting the exposure of gradient information from passive clients.
>
> However, we would like to clarify that **privacy concerns are beyond the main focus of this paper**. Our main focus is on addressing the challenges of online learning within VFL.

---

> ### Author Response · Authors · 2024-11-20
> **Response to Reviewer H7dv - Part 3**
>
> > *Q3: How does the framework's performance scale with different total numbers of clients?*
>
> Thank you for your suggestion. To address this, we extended our main experiments in Section 5.2 to include scenarios with **8 and 16 clients**. The corresponding results are available via this anonymous [\[clickable link\]](https://anonymous.4open.science/r/EventDrivenOnlineVFL_ICLR_Disucssion-2748/iMNIST/iMNIST_IID_client8_subfig.png).
> These experimental results will be added to the appendix. The conclusion remains consistent with our previous findings: OGD is less stable under partial client activation, while DLR converges more rapidly than SLR. This consistency supports the robustness and scalability of our proposed framework.

---

> ### Author Response · Authors · 2024-11-25
> **Follow-Up on Reviewer Feedback and Submission Updates**
>
> **Thank you very much for taking the time to review our paper. We sincerely appreciate your valuable feedback and have carefully addressed the concerns in our previous response. Does our response address your concerns?**
>
>
> Besides, we have updated the submission. The experiment results in **Q3** (Scalability) and **W4** (# Activated Clients) have been added to **Appendix D.3**, while the table in **Q1** (Regret Comparison) has been included in **Appendix E.2**.

---

> > ### Comment · Reviewer_H7dv · 2024-11-27
> >
> > Thank you for the response, I will raise my score accordingly.

---

> > > ### Author Response · Authors · 2024-11-27
> > > **Thank you!**
> > >
> > > Thank you so much for taking the time to thoroughly review our paper and for providing thoughtful comments that have helped us improve our work.
> > >
> > > We deeply appreciate your support and valuable feedback!

---

### Comment · Area_Chair_wfAV · 2024-11-25
**Acknowledge the author responses**

Dear Reviewers,

Thank you very much for your effort. As the discussion period is coming to an end, please acknowledge the author responses and adjust the rating if necessary.

Sincerely,
AC

---

> ### Author Response · Authors · 2024-11-28
> **Summary of Revisions and Clarifications**
>
> We sincerely thank the reviewers for their valuable time and effort in thoroughly reviewing our paper and providing thoughtful feedbacks. We have revised our submission accordingly.
>
> Below, we present a summary of the improvements and revisions made during the discussion phase compared to our original submission.
>
> | **Location**| **Content**                                             | **Discussion with Reviewer**         |
> |----------------------------------------|----------------------------------------------------------------------|---------------------------------------|
> |**Figures and Tables**                |                             |
> | Appendix D.2                           | SLR lines for SUSY and HIGGS experiment                              | qrpn (Q1)                             |
> | Appendix D.3                           | Experiment on Scalability of the framework (8 and 16 clients)        | H7dv (Q3), MnZ6 (W2)                  |
> | Appendix D.3                           | Experiment on Number of Activated Clients                            | H7dv (W4)                             |
> | Appendix D.3                           |Table of Client Activation Statistics                                         | qrpn (Follow-up questions)            |
> | Appendix E.2                           |Table of Regret Bound Comparison                                  | H7dv (Q1)                             |
> | **Typos and Illustration Clarity**      |                                                                      |                                       |
> | #146, Eq. 1                            | $f^t(w_0, w, x^t, y^t)$, illustration clarity                         | x1ux (Q1)                             |
> | #164                                   | $\frac{1}{M} $ $\rightarrow$ $\frac{1}{W} $, typo                                                    | MnZ6 (Q3)                             |
> | #278                                   | $\nabla F^t $ $\rightarrow$  $\nabla f^t$, typo                                                    | MnZ6 (Q7)                             |

---

### Meta-Review · Area_Chair_wfAV · 2024-12-19

**Metareview:**

This paper proposes an event-driven online vertical federated learning (VFL) framework.  The reviewers agreed that the paper tackles an interesting problem and the solution is well-designed and well-presented.  Also, the reviewers raised several concerns mostly on theoretical analysis and evaluation.  The authors successfully addressed many of the concerns during the discussion period.  Overall, since all the reviewers are positive on the acceptance of this paper, I am happy to recommend an accept.

**Additional Comments On Reviewer Discussion:**

All the reviewers were satisfied with the authors' responses during the discussion period.

---

### Decision · Program_Chairs · 2025-01-22

Accept (Poster)